# SIMSHIFT: A Benchmark for Adapting Neural Surrogates to Distribution Shifts

## Abstract

Neural surrogates for Partial Differential Equations (PDEs) often suffer significant performance degradation when evaluated on unseen problem configurations, such as novel material types or structural dimensions. Meanwhile, Domain Adaptation (DA) techniques have been widely used in vision and language processing to generalize from limited information about unseen configurations. In this work, we address this gap through two focused contributions. First, we introduce SIMSHIFT, a novel benchmark dataset and evaluation suite composed of four industrial simulation tasks: *hot rolling*, *sheet metal forming*, *electric motor design* and *heatsink design*. Second, we extend established domain adaptation methods to state of the art neural surrogates and systematically evaluate them. These approaches use parametric descriptions and ground truth simulations from multiple source configurations, together with only parametric descriptions from target configurations. The goal is to accurately predict target simulations without access to ground truth simulation data. Extensive experiments on SIMSHIFT highlight the challenges of out of distribution neural surrogate modeling, demonstrate the potential of DA in simulation, and reveal open problems in achieving robust neural surrogates under distribution shifts in industrially relevant scenarios.

## 1 Introduction

Simulations based on PDEs are essential tools for understanding and predicting physical phenomena in engineering and science [1]. Over recent years, machine learning has emerged as a promising and novel modeling option for complex systems [2], significantly accelerating and augmenting simulation workflows across diverse applications, including weather and climate forecasting [3, 4, 5, 6], material design [7, 8, 9] and protein folding [10, 11], amongst others.

In practice, however, models are often deployed in settings where simulation configurations differ from those seen during training. This *distribution shift* [12] often leads to significant degradation in performance [13, 14, 15], making reliable deployment of neural surrogates in industrial workflows less likely. Some industry relevant studies propose post simulation correction [16], identify limited parameter variation as a constraint [17], or consider out of distribution tasks without tailored solutions [13].

While methods for increasing out of distribution performance have been at the center of research for a long time [12, 18, 19, 20, 21, 22, 23], to the best of our knowledge, no benchmark systematically investigates such methods on simulation tasks [24, 25, 26, 13, 17, 27, 28, 29]. Addressing this gap is particularly relevant in scientific and industrial settings, where generating ground truth simulation data is costly and limits the diversity of training configurations. In contrast, parametric descriptions, such as material types or structural dimensions, are often readily available or easy to generate.

Submitted to 39th Conference on Neural Information Processing Systems Datasets and Benchmarks Track (NeurIPS D&B 2025). Do not distribute.

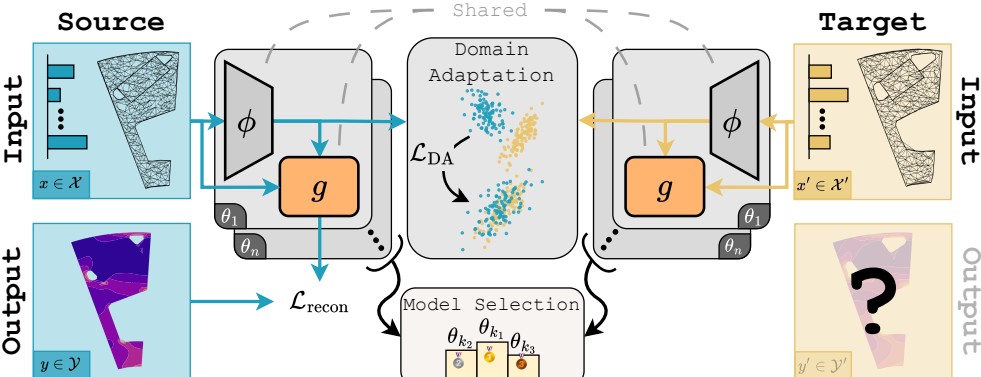

Figure 1: Schematic overview of the SIMSHIFT framework. During model training, we have access to inputs (e.g. parameters and meshes) and corredsponding outputs $(x, y)$ from the source domain (left, blue), and only inputs $x'$ from the target domain (right, yellow). The neural operator $g$ and the conditioning network $\phi$ are shared across domains and jointly optimized. Models are trained with two loss terms, namely $\mathcal{L}_{\text{recon}}$, which is computed on source labels, and $\mathcal{L}_{\text{DA}}$, which aligns source and target conditioning features. After training, unsupervised model selection strategies choose the model $\theta_{k1}$ expected to perform best on the target domain.

This problem is known as *Unsupervised Domain Adaptation (UDA)* [30], where parametric (input) descriptions and full simulation outputs are available for each *source* configuration, while only input descriptions are provided for *target* configurations, without corresponding outputs. Decades of UDA research have produced effective methods for addressing domain gaps [31, 32, 33], yet their potential for PDE surrogate modeling remains largely unexplored.

To investigate the potential of UDA for neural surrogate modeling, we provide simulation data from diverse simulation configurations, across a range of realistic tasks from engineering design. Our settings are all rooted in application and derived from industrial problem settings. We introduce a comprehensive benchmark that evaluates established UDA methods and neural surrogates. An overview of the framework is shown in Figure 1. Our contributions can be summarized as follows:

- We propose four practical datasets with predefined distribution shifts in *hot rolling*, *sheet metal forming*, *electric motor*, and *heatsink* design, based on realistic simulation setups.

- We present, to the best of our knowledge, the first joint study of established neural surrogate architectures and UDA on engineering simulations with unstructured meshes.

- We introduce *SIMSHIFT*, a modular benchmarking suite that complements our datasets with baseline models and algorithms. It allows easy integration of new simulations, machine learning methods, domain adaptation techniques, and model selection strategies.

## 2   Related Work

**Unsupervised Domain Adaptation.** UDA research covers a wide spectrum of results from theoretical foundations [18, 34, 30, 35] to modern deep learning methods [36, 37, 38, 23, 39, 40, 41, 42, 43, 44, 45]. A prominent class of methods, dubbed as *representation learning*, aims to map the data to a feature space, where source and target representations appear similar, while maintaining enough information for accurate prediction. To enforce feature similarity between domains, algorithms often employ statistical [46, 47, 23, 48, 49, 50, 51, 52, 53, 54] or adversarial [22, 55] discrepancy measures. One crucial yet frequently overlooked factor in the success of UDA methods is model selection. Numerous studies underline the critical impact of hyperparameter choices on UDA algorithm performance, often overshadowing the adaptation method itself [56, 32, 57, 58, 59]. Even more, since labeled data is unavailable in the target domain, standard validation approaches (including validation sets, ensembling or information criteria) become infeasible. Thus, it is essential to jointly evaluate adaptation algorithms alongside their associated unsupervised model selection strategies. In

this work, we focus on importance weighting strategies [60, 61, 58], which stand out by their general applicability, theoretical guarantees and high empirical performance.

**Benchmarks for UDA.** Numerous benchmark datasets and evaluation protocols have been established for UDA methods across various machine learning domains, including computer vision [62, 63, 64, 65, 66], natural language processing [67], timeseries data [68] and tabular data [69]. However, to the best of our knowledge, systematic UDA benchmarking for neural surrogates remains unexplored.

**Neural Surrogates.** One prominent approach in neural surrogate modeling for PDEs is operator learning [70, 71, 72, 73, 74]. In this setting, an operator maps input functions, such as boundary or initial conditions, to the corresponding solution of the PDE. During training, neural operators typically learn from input-output pairs of discretized functions [70, 71, 72, 73]. While some methods expect regular, grid based inputs [71], others can be applied to any kind of data structure [73, 74]. One notable property is *discretization invariance*, which, along with the ability to handle irregular data, enables generalization across different resolutions and mesh geometries. This is a highly desirable property for industrial simulations [75, 73, 76, 77, 78], where non-uniform meshes are the standard due to the computational and modeling advantages. In this work, we focus on domain adaptation rather than benchmarking discretization invariance, and include neural surrogates that may not satisfy this property, such as [79].

**Benchmarks for Neural Surrogates.** Benchmarks for neural surrogates have made substantial progress, providing new datasets and metrics specific to PDE problems. Many focus on solving PDEs on structured, regular grids [24, 25, 26], which serve as valuable platforms for developing and testing new algorithms. However, these overlook the irregular meshes commonly used in large scale industrial simulations. In that direction, other benchmarks extend to Computational Fluid Dynamics (CFD) on irregular static meshes for airfoil simulations [13], aereodynamics for automotive [17, 27], more traditional fluid study problems [28], and even particle based Smoothed Particle Hydrodynamics simulations [29, 80]. Finally, and most closely related to our work, recent efforts have explored the application of Active Learning techniques [81, 82] to neural surrogates, introducing a benchmark specifically designed for data-scarce scenarios [83].

Despite these contributions, all current benchmarks often fall short when addressing a critical issue: the significant performance drop learned models exhibit under distribution shifts, i.e., when encountering simulation configurations beyond their training setting [12].

# 3 Dataset Presentation

Our datasets follow three design principles. (i) Industry relevance: They reflect a practical, real-world simulation use-case. The benchmark covers a diverse set of problems, including 2D as well as 3D cases. (ii) Parametrized conditions: The behavior of all simulations depends on the set of initial parameters only. (iii) steady state scenarios: We constrain them to time independent problems, in order to avoid additional complexity such as autoregressive error accumulation in neural surrogates [84].

The datasets were generated using the commercial Finite Element Method (FEM) software *Abaqus*[1], the open-source simulation software *HOTINT*[2] and the open-source CFD package *OpenFoam 9*[3]. An overview of each dataset is presented in Sections 3.1 to 3.4. Additionally, we present detailed descriptions of the respective numerical simulations provided in the technical supplementary material.

Since the behavior of each simulation task is entirely determined by its input parameters, we predefine source and target domains by partitioning the parameter space into distinct, non-overlapping regions. A detailed explanation of the domain splitting strategy is provided in Section 3.5.

Each dataset includes three levels of distribution shift difficulty: *easy*, *medium* and *hard*. These levels reflect increasing domain gap magnitudes in parameter space. In this work, we benchmark the *medium* difficulty for each dataset and, for clarity, provide error scaling results across all levels for the *hot rolling* dataset (Figure 6).

---

[1]https://www.3ds.com/products/simulia/abaqus
[2]https://hotint.lcm.at/
[3]https://www.openfoam.com/

In total, we collect four datasets leading to 12 domain adaptation tasks. Table 1 summarizes key characteristics of each dataset, including physical dimensionality, mesh resolution, number of conditioning parameters, and total dataset size. All datasets are publicly hosted on Hugging Face[4] for convenient access.

Table 1: Overview of the benchmark datasets. The heatsink meshes were subsampled to a fourth of their original size during preprocessing. For a detailed description the simulation parameter sampling ranges, see Appendix E.

| Dataset | Origin | Samples | Output channels | Avg. # nodes | Varied simulation parameters | Dim | (GB) |
|---------|--------|---------|-----------------|--------------|------------------------------|-----|------|
| Rolling | Metallurgy | 4,750 | 10 | 576 | 4 | 2D | 0.5 |
| Forming | Manufacturing | 3,315 | 10 | 6,417 | 4 | 2D | 4.1 |
| Motor | Machinery | 3,196 | 26 | 9,052 | 15 | 2D | 13.4 |
| Heatsink | Electronics | 460 | 5 | 1,385,594 | 4 | 3D | 40.8 |

## 3.1 Hot Rolling

The rolling dataset captures a *hot rolling* process, where a metal slab is plastically deformed into a sheet metal product, as visualized in Figure 2. This complex thermo-mechanical operation involves tightly coupled elasto-plastic deformation and heat transfer phenomena [85, 86, 87]. The Finite Element simulation models the progressive thickness reduction and thermal evolution of the material as it passes through a rolling gap, incorporating temperature-dependent material properties and contact between the slab and the rolls.

Key input parameters include the initial slab thickness $t$, temperature characteristics $T_{\text{core}}$ and $T_{\text{surf}}$ of the slab, as well as the geometry of the roll gap. To vary the slab deformation we define the thickness reduction as a percentage of the initial thickness: reduction $= \frac{t-g}{t}$, where $g$ is the rolling gap distance. Table 10 in Appendix E.1 shows a detailed overview of the parameter values together with their sampling ranges used to generate the dataset.

The 2D simulation outputs various field quantities, with the most important being Equivalent Plastic Strain (PEEQ), a scalar field representing the materials plastic deformation, shown in Figure 2b.

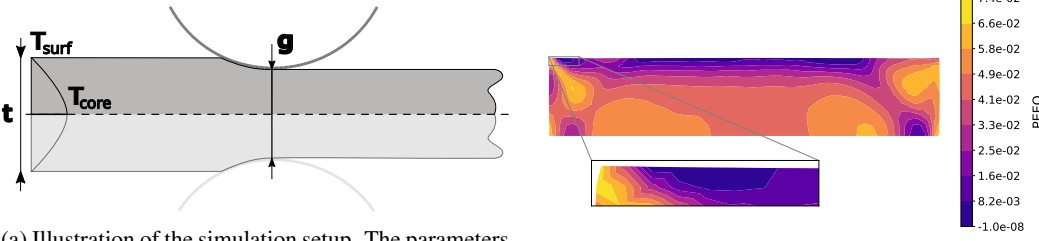

(a) Illustration of the simulation setup. The parameters correspond to those in Table 10. We use symmetry constraints and only simulate one half of the slab.

(b) Metal slab after the process, showing PEEQ as a contour plot.

Figure 2: Overview of the *hot rolling* simulation scenario.

## 3.2 Sheet Metal Forming

The forming dataset represents a *sheet metal forming* process, a critical manufacturing operation widely used across industries such as automotive, aerospace, and industrial equipment manufacturing. FEM simulations are commonly employed to estimate critical quantities such as thinning, local plastic deformation and residual stress distribution with high accuracy [88, 89, 90].

The simulated setup in this dataset consists of a symmetrical sheet metal workpiece supported at the ends and center, a holder and a punch that deforms the sheet by applying a displacement denoted

[4]https://huggingface.co/datasets/simshift/SIMSHIFT_data

as $U$. Figure 3a visualizes the process. During the process, the metal sheet undergoes elasto-plastic deformation, transitioning from a flat initial state to a "w-shaped" geometry.

Variable input parameters include half the deformed sheet length $l$, the sheet thickness $t$, friction coefficient $\mu$ and the radii of the holder, punch, and supports $r$. Table 11 in Appendix E.2 provides the sampling ranges for data generation. The 2D model simulates the forming procedure and predicts the sheet's deformation behavior, providing field quantities such as stress, as well as elastic and plastic strain distributions, one of which is shown in Figure 3b.

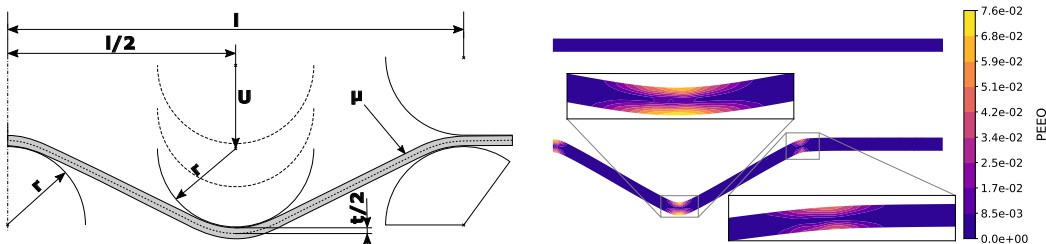

(a) Illustration of the simulation setup. The parameters correspond to those listed in Table 11.

(b) Material before (top) and after (bottom) the process, showing the PEEQ field as a contour plot.

Figure 3: Overview of the *sheet metal forming* simulation scenario.

## 3.3 Electric Motor Design

The electric motor dataset encompasses a structural FEM simulation of a rotor in electric machinery, subjected to mechanical loading at burst speed. This simulation is motivated by the inherently conflicting design objectives in rotor development: while magnetic performance favors certain rotor topologies to optimize flux paths and torque generation, structural integrity requires designs capable of withstanding centrifugal loads without plastic deformation [91, 92]. The simulation predicts stress and deformation responses due to assembly pressing forces and centrifugal loads, accounting for the rotor's topology, material properties, and rotation speed.

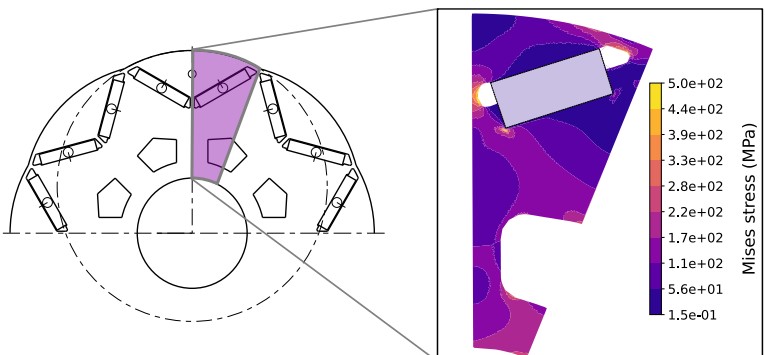

Figure 4: The *electric motor design* simulation scenario, with a schematic sketch of the motor (left) and zoomed-in detail from the simulated radial portion (right). Mises stress field contour plot is shown.

Figure 4 shows an overview of the simulation setup. Since this case is more complex than the preceding datasets, we omit a detailed technical drawing from the main body and instead provide it in Figure 11, besides the corresponding parameter variations in Table 12, both in Appendix E.3.

## 3.4 Heatsink Design

The heat sink dataset represents a CFD simulation focused on the thermal performance of heat sinks, commonly used in electronic cooling applications [93, 94].

It models the convective heat transfer from a heated base through an array of fins to the surrounding air. The simulation captures how geometric fin characteristics, specifically, the number, height, and thickness of fins, affect the overall heat dissipation, along with the temperature of the heat sink.

The 3D CFD model outputs include steady state temperature (see Figure 5), velocity and pressure fields, enabling the assessment of design efficiency and thermal resistance under varying configurations. An overview of the setup as well as key parameters are provided in Appendix E.4.

### 3.5 Distribution Shifts

To define distribution shifts of varying difficulties and corresponding source and target domains, we focus on the most influential input parameter in each simulation scenario, which is identified by domain experts. To further validate the opinions of the experts, we perform clustering analyses on the latent representations of models trained across the full parameter range. In general, the resulting clusters confirm the sensitivity of the latent space to the chosen dominant parameter. Visualizations of t-SNE plots of the latent spaces with the respective clusters are provided in Figures 7 to 10. The chosen parameters and their respective ranges for the different domains are provided in Table 7.

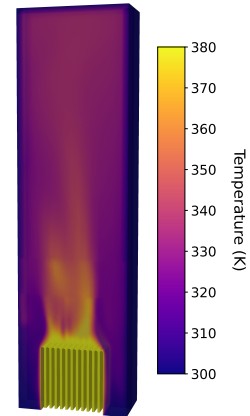

Figure 5: Sliced view of the temperature field of a *heatsink design* simulation.

## 4 Benchmark Setup

This section outlines the learning problem (Section 4.1), the domain adaptation algorithms considered (Section 4.2), the unsupervised model selection strategies (Section 4.3), and the baseline models used (Section 4.4). Finally, we describe the experimental setup and evaluation metrics in Section 4.5.

### 4.1 Learning Problem

Let $\mathcal{X}$ be an input space $\mathcal{X}$ containing geometries and conditioning parameters (e.g., thickness and temperatures in Figure 2a) and $\mathcal{Y}$ be an output space containing ground truth solution fields obtained from a numerical solver (e.g., PEEQ field in Figure 2b). Following [30], a *domain* is represented by a probability density function $p$ on $\mathcal{X} \times \mathcal{Y}$ (e.g., describing the probability of observing an input-output pair corresponding to the parameter range $r \in [0.01, 0.115)$ in Table 7). UDA has been formulated as follows: Given a source dataset $(x_1, y_1), ..., (x_n, y_n)$ drawn from a source domain $p_S$ together with an *unlabeled* target dataset $x'_1, ..., x'_m$ drawn from the ($\mathcal{X}$-marginal) of a target domain $p_T$, the problem is to find a model $f : \mathcal{X} \to \mathcal{Y}$ that has small expected risk on the target domain:

$$\mathbb{E}_{(x,y)\sim p_T}\left[\ell(f(x), y)\right], \qquad (1)$$

with $\ell : \mathcal{Y} \times \mathcal{Y} \to \mathbb{R}$ being some loss function. For example, consider the square loss $\ell(f(x), y) = (f(x) - y)^2$ and Figure 1, where $f(x) = g(x, \phi(x))$ is composed of a conditioning network $\phi$ and a surrogate $g$.

### 4.2 Unsupervised Domain Adaptation Algorithms

Our UDA baseline algorithms are from the class of *domain-invariant representation learning* methods. These methods are strong baselines, in the sense that their performance typically lies within the standard deviation of the winning algorithms in large scale empirical evaluations (i.e., no significant outperformance is observed), see CMD, Deep-CORAL and DANN in [58, Tables 12–14], M3SDA in [95], MMDA and HoMM in [68].

Following [49, 57], we express the objective of domain-invariant learning using two learning models: a *representation* mapping $\phi \in \Phi \subset \{\phi : \mathcal{X} \to \mathcal{R}\}$, which in our case corresponds to the conditioning network that maps simulation parameters into some representation space $\mathcal{R} \subset \mathbb{R}^m$ and a *regressor* $g \in \mathcal{G} \subset \{g : \mathcal{X} \times \mathcal{R} \to \mathcal{Y}\}$, which is realized by a neural surrogate. The goal is to find a mapping $\phi$ under which the source representations $\phi(\mathbf{x}) := (\phi(x_1), \ldots, \phi(x_n))$ and the target representations

$\phi(\mathbf{x}') := (\phi(x_1'), \dots, \phi(x_m'))$ appear similar, and, at the same time, enough information is preserved for prediction by $g$, see [12]. This is realized by estimating objectives of the form

$$\min_{g \in \mathcal{G}, \phi \in \Phi} \mathbb{E}_{(x,y) \sim p_T} [\ell(g(x, \phi(x)), y)] + \lambda \cdot d(\phi(\mathbf{x}), \phi(\mathbf{x}')), \qquad (2)$$

where $d$ is a distance between source and target representations and $\lambda$ is a regularization parameter. Good choices for $d$ in Eq. (2) have been found to be the Wasserstein distance [53, 54], the Maximum Mean Discrepancy [51, 52], moment distances [46, 23], adversarially learned distances [22, 55] and other measures of divergence [48, 49, 50]. Appropriately choosing $\lambda$ is crucial for high performance [56, 32, 58, 61, 59], making model selection necessary.

### 4.3 Unsupervised Model Selection

Among all algorithm design choices in UDA, model selection has been repeatedly recognized as one of the most crucial [56, 32, 58, 61, 59], with sub-optimal choices potentially leading to *negative transfer* [33]. However, classical approaches (e.g., validation set, cross-validation, information criterion) cannot be used due to missing labels and distribution shifts. It is therefore a natural benchmark requirement for UDA to provide also unified model selection strategies in addition to UDA algorithms.

In this work, we rely on Importance Weighted Validation (IWV) [60] and Deep Embedded Validation (DEV) [61] to overcome the two challenges: (i) distribution shift and (ii) missing target labels. These methods rely on the Radon-Nikodým derivative and the covariate shift assumption $p_S(y|x) = p_T(y|x)$ to obtain

$$\mathbb{E}_{(x,y) \sim p_T}[\ell(f(x), y)] = \mathbb{E}_{(x,y) \sim p_S}\left[\frac{p_T(x)\cancel{p_T(y|x)}}{p_S(x)\cancel{p_S(y|x)}}\ell(f(x), y)\right] = \mathbb{E}_{(x,y) \sim p_S}[\beta(x)\ell(f(x), y)] . \quad (3)$$

Eq. (3) motivates to estimate the target error by a two step procedure: First, approaching challenge (i) by estimating the density ratio $\beta(x) = \frac{p_T(x)}{p_S(x)}$ from the input data only, and, approaching challenge (ii) by estimating the target error by the weighted source error using the *labeled* source data.

### 4.4 Baseline Models

We provide a comprehensive range of machine learning methods, adapted to our conditioned simulation task, organized by their capacity to model interactions across different spatial scales:

*Global context models* such as PointNet [96] incorporate global information into local Multi-Layer Perceptrons (MLPs) by summarizing features of all input points by aggregation into a global representation, which is then shared among nodes. Recognizing the necessity of *local information* when dealing with complex meshes and structures, we include GraphSAGE [79], a proven Graph Neural Network (GNN) architecture [97, 98] already used in other mesh based tasks [75, 13]. However, large scale applications of GNNs are challenging due to computational expense [73] and issues like oversmoothing [99]. Finally, to overcome these limitations, we employ *attention based models* [100]. These models typically scale better with the number of points, and integrate both global and local information enabling stronger long-range interactions and greater expressivity. We include Transolver [101], a modern neural operator Transformer.

As an alternative categorization, baselines can also be classified by input-output pairings into *point-to-point* and *latent* approaches. The former explicitly encodes nodes, while the latter represents the underlying fields in a latent space and requires queries to retrieve nodes. All previously mentioned models are *point-to-point*, and as an example of a latent field method, we include Universal Physics Transformer (UPT) [73, 76] . UPTs are designed for large scale problems and offer favorable scaling on large meshes through latent field modeling; however they are better suited for static-mesh scenarios, as they are lacking the notion of point and don't handle deformations out-of-the-box. Therefore we benchmark this approach only on the *heatsink design* dataset.

Finally, all our tasks require neural operators to be explicitly conditioned on configuration parameters of the numerical simulations. To achieve this, we embed these parameters using an embedding and a shallow MLP (denoted as $\phi$ in Section 4.2 and Figure 1) to produce a latent representation. Subsequently, we condition the neural operator using either concatenation of this latent conditioning vector with the global features, or scale-shift modulation of intermediate features using FiLM or DiT

conditioning layers [102, 103]. Detailed explanations of all implemented architectures are given in Appendix C.

## 4.5 Experiments and Evaluation

**Experimental Setup.** We benchmark the three prominent UDA algorithms Deep-Coral [46], CMD [23] and DANN [22], in combination with the four unsupervised model selection strategies IWV [60], DEV [61], Source Best (SB), which selects models based on source domain validation performance, and, Target Best (TB), which is the (oracle) best performing model (over all runs with all hyperparameters) that is selected by hand using the target simulation data (that is not available in UDA).

For the baseline neural surrogate models, we evaluate PointNet [96], GraphSAGE [79], and Transolver [101] on the *hot rolling*, *sheet metal forming*, and *electric motor design* datasets. Due to memory and runtime constraints on the large scale *heatsink design* dataset, we omit GraphSAGE and instead benchmark UPT [73] alongside PointNet and Transolver.

**Experimental Scale.** In total, this results in $3_{\text{models}} \times 3_{\text{UDA algorithms}} \times 4_{\text{selection algorithms}} + 3_{\text{unregularized models}} = 39$ configurations per dataset (i.e. number of lines per results table in Appendix A). We perform an extensive sweep over the critical UDA parameter $\lambda$ and average across four seeds, totaling in $1,200$ training runs.

Full details on architectures, hyperparameters, training setup and normalization, as well as a breakdown of training times are included in Appendices C and D.

**Evaluation Metrics.** For each dataset, we report the averaged Root Mean Squared Error (RMSE) over all normalized output fields, as well as the averaged per field RMSE values (computed on denormalized data) and the Euclidean error for deformation predictions. Detailed metric definitions are provided in Appendix D.2.

# 5 Benchmarking Results

Table 2 presents an overview of the benchmarking results. Overall, we observe consistent improvements in target domain performance with the application of UDA algorithms and unsupervised model selection strategies, validating their effectiveness.

While the results in Table 2 suggest a minor performance decline on the *Forming* dataset, this is not representative of the full performance across all output fields. As only selected outputs are shown

Table 2: Best performing UDA algorithm & unsupervised model selection combination for all model architectures across all datasets. Additionally, we provide an oracle (TB), which demonstrates the theoretical lower bound on error. Values show the denormalized average RMSE per field in the target domain. Differences to the model trained without UDA are shown in parentheses, where negative values indicate performance improvements. Dashes (–) indicate fields not present in the respective dataset. The best performing models were chosen based on the average RMSE across all normalized fields of the respective datasets (see detailed results in Appendix A).

| Dataset | All Models | Best UDA method | Best model selection | Deformation (mm) | Mises stress (MPa) | Equivalent plastic strain ($\times 10^{-2}$) | Temperature (K) | Velocity (m/s) |
|---|---|---|---|---|---|---|---|---|
| Rolling | PointNet | CMD | SB | 11.33 (-0.15) | 27.92 (+0.31) | 2.51 (-0.01) | – | – |
| | GraphSAGE | CMD | IWV | **4.62 (-1.09)** | **14.49 (-5.30)** | **1.56 (-0.55)** | – | – |
| | Transolver | CMD | SB | 13.87 (-579.11) | 77.74 (-6409.53) | 5.80 (-126.88) | – | – |
| | Oracle (GraphSAGE) | Deep Coral | TB | 4.55 (-1.17) | 13.83 (-5.96) | 1.43 (-0.69) | – | – |
| Forming | PointNet | Deep Coral | SB | 2.56 (-0.00) | 31.35 (-0.09) | **0.15 (-0.01)** | – | – |
| | GraphSAGE | DANN | IWV | 2.10 (+0.16) | 52.40 (+6.30) | 0.27 (-0.00) | – | – |
| | Transolver | Deep Coral | DEV | **1.39 (+0.20)** | 25.05 (+2.04) | **0.15 (+0.02)** | – | – |
| | Oracle (Transolver) | CMD | TB | 1.02 (-0.17) | 20.28 (-2.73) | 0.12 (-0.01) | – | – |
| Motor | PointNet | Deep Coral | SB | 1.53 (-0.06) | 26.23 (-4.43) | – | – | – |
| | GraphSAGE | CMD | SB | 1.31 (-0.19) | 28.92 (-0.54) | – | – | – |
| | Transolver | Deep Coral | SB | **1.30 (-0.20)** | **7.68 (-0.65)** | – | – | – |
| | Oracle (Transolver) | Deep Coral | TB | 1.25 (-0.24) | 7.59 (-0.73) | – | – | – |
| Heatsink | PointNet | Deep Coral | SB | – | – | – | 17.43 (-3.70) | 0.044 (+0.000) |
| | Transolver | Deep Coral | IWV | – | – | – | 13.43 (+0.00) | 0.041 (+0.001) |
| | UPT | Deep Coral | SB | – | – | – | **12.41 (-0.62)** | **0.039 (-0.001)** |
| | Oracle (UPT) | Deep Coral | TB | – | – | – | 12.64 (-0.40) | 0.039 (-0.001) |

here, the observed gains in other fields captured by the mean normalized RMSE are not visible in this summary (see Table 4).

Despite the clear benefits provided by UDA, we find that no single UDA algorithm or unsupervised model selection strategy consistently outperforms the others across all datasets. Furthermore, the evident gap between the best performing UDA algorithms and model selection strategies compared to the theoretical lower bound provided by the Target Best (TB) oracle indicates that existing unsupervised model selection strategies still leave substantial room for improvement.

Finally, since the presented tables only report performance on the *medium* difficulty setting, we additionally visualize model behavior of the best performing combination (model + UDA algorithm + selection strategy: *CMD + IWV*) across all difficulty levels of the *hot rolling* dataset in Figure 6. It illustrates the increase in prediction error as the domain gap widens and highlights the consistent improvements achieved by applying UDA algorithms combined with unsupervised model selection strategies on the *easy* and *medium* settings.

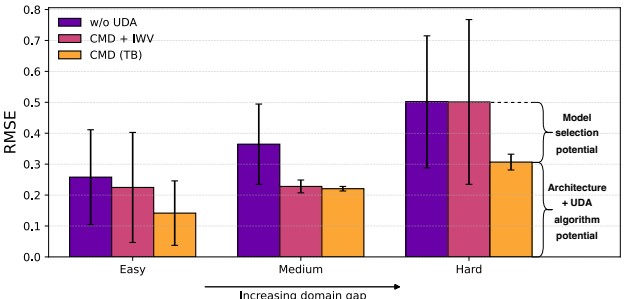

Figure 6: Error scaling with increasing domain gap. We show the averaged RMSE across all (normalized) fields for the *easy*, *medium*, and *hard* gaps on the *hot rolling* task. We compare models without UDA, the best performing UDA method with unsupervised model selection (CMD + IWV), and the theoretical lower bound (TB). Error bars indicate the standard deviation across 4 seeds. Furthermore, we highlight potentials of selection improvements on the *hard*.

For the *hard* setting, however, the shown unsupervised model selection algorithm fails to identify suitable models, as the mean error matches that of the unregularized baselines with the standard deviation even increasing. Nonetheless, the theoretical lower bound (TB) remains substantially below the unregularized error. This indicates the two promising directions for further improvement of the presented baselines: (i) enhancement of neural surrogate architectures and UDA algorithms, and (ii) especially, improvement of unsupervised model selection strategies.

## 6  Discussion

We presented SIMSHIFT, a collection of industry relevant datasets paired with a benchmarking library for comparing UDA algorithms, unsupervised model selection strategies and neural operators in real word scenarios. We adapt available techniques and apply them on physical simulation data and perform extensive experiments to evaluate their performance on the presented datasets. Our findings suggest that standard UDA training methods can improve performance of neural operators to unseen parameter ranges in physical simulations, with improvement margins in line with those seen in UDA literature [58, 68]. Additionally, we find correct unsupervised model selection to be extremely important in downstream model performance on target domains, with it arguably having as much impact as the UDA training itself, which is also in agreement with other DA works [56].

**Limitations.** We acknowledge that our datasets are limited under three main aspects: (i) They only cover *steady state* problems, whereas there is a growing interest in modeling *time dependent* PDEs with neural operators. (ii) By defining domains with parameter ranges, we restrict the shifts to *"scalar"* gaps, disregarding changes in mesh geometry (e.g. topology or geometric transformations). (iii) The defined domain shifts currently emphasize variations in a single parameter rather than exploring more realistic shifts involving multiple parameters simultaneously. These three choices are motivated by considering benchmarking simplicity and computational constraints, and are open for future extensions.

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

# Appendix

# A  Detailed results

## A.1  Hot Rolling

Table 3: Mean (± standard deviation) of RMSE across four seeds on the *hot rolling* dataset. Bold values indicate the best target domain performance across all normalized fields. Underlined entries mark the best performing UDA algorithm and unsupervised model selection strategy per model. Asterisks denote unstable models (error more than 10× higher than others).

| Model | DA Algorithm | Model Selection | All Fields normalized avg (-) SRC | TGT | Deformation (mm) SRC | TGT | Logarithmic strain (×10⁻²) SRC | TGT | Equivalent plastic strain (×10⁻²) SRC | TGT | Mises stress (MPa) SRC | TGT | Stress (MPa) SRC | TGT |
|---|---|---|---|---|---|---|---|---|---|---|---|---|---|---|
| | - | - | 0.016(±0.000) | 0.365(±0.130) | 0.525(±0.023) | 5.715(±1.567) | 0.018(±0.000) | 0.997(±0.377) | 0.033(±0.000) | 2.113(±0.789) | 1.972(±0.024) | 19.790(±7.186) | 1.234(±0.010) | 11.421(±3.891) |
| GraphSAGE | DANN | DEV | 0.014(±0.000) | 1.175(±0.053) | 0.577(±0.061) | 17.363(±0.803) | 0.019(±0.001) | 3.452(±0.176) | 0.035(±0.001) | 7.290(±0.405) | 2.056(±0.050) | 111.626(±7.317) | 1.264(±0.033) | 59.263(±5.594) |
| | DANN | IWV | 0.014(±0.000) | 0.289(±0.147) | 0.561(±0.032) | 5.359(±1.848) | 0.018(±0.001) | 0.792(±0.186) | 0.033(±0.001) | 1.622(±0.306) | 1.992(±0.037) | 24.471(±22.423) | 1.246(±0.025) | 13.737(±11.828) |
| | DANN | SB | 0.014(±0.000) | 0.692(±0.511) | 0.573(±0.043) | 11.090(±7.161) | 0.018(±0.000) | 2.120(±1.506) | 0.034(±0.001) | 4.510(±3.201) | 1.991(±0.045) | 60.352(±51.358) | 1.237(±0.027) | 31.612(±25.882) |
| | DANN | TB | 0.014(±0.000) | 0.230(±0.041) | 0.604(±0.010) | 4.640(±0.593) | 0.018(±0.001) | 0.740(±0.134) | 0.034(±0.001) | 1.549(±0.275) | 2.017(±0.047) | 14.867(±3.085) | 1.248(±0.028) | 8.665(±1.635) |
| | CMD | DEV | 0.015(±0.001) | 1.447(±0.202) | 0.617(±0.040) | 18.383(±2.116) | 0.020(±0.001) | 3.781(±0.544) | 0.037(±0.003) | 7.764(±1.210) | 2.169(±0.151) | 136.324(±23.104) | 1.324(±0.062) | 95.502(±20.973) |
| | CMD | IWV | 0.014(±0.000) | **0.228(±0.021)** | 0.577(±0.023) | 4.622(±0.283) | 0.018(±0.000) | 0.742(±0.071) | 0.033(±0.001) | 1.563(±0.153) | 1.980(±0.040) | 14.494(±1.375) | 1.237(±0.032) | 8.386(±0.819) |
| | CMD | SB | 0.014(±0.000) | 0.786(±0.535) | 0.571(±0.040) | 12.160(±7.174) | 0.018(±0.000) | 2.403(±1.541) | 0.033(±0.000) | 5.068(±3.248) | 1.974(±0.034) | 68.509(±55.017) | 1.228(±0.024) | 36.834(±28.921) |
| | CMD | TB | 0.014(±0.000) | 0.221(±0.007) | 0.583(±0.033) | 4.607(±0.261) | 0.018(±0.000) | 0.711(±0.014) | 0.033(±0.000) | 1.507(±0.045) | 1.992(±0.021) | 14.288(±1.040) | 1.245(±0.019) | 8.275(±0.632) |
| | Deep Coral | DEV | 0.014(±0.000) | 0.668(±0.351) | 0.519(±0.066) | 10.566(±4.767) | 0.018(±0.000) | 2.042(±1.017) | 0.033(±0.000) | 4.291(±2.145) | 1.992(±0.014) | 56.626(±37.354) | 1.241(±0.008) | 30.864(±18.487) |
| | Deep Coral | IWV | 0.014(±0.000) | 0.282(±0.056) | 0.569(±0.040) | 5.416(±0.356) | 0.018(±0.000) | 0.874(±0.103) | 0.033(±0.000) | 1.841(±0.199) | 1.977(±0.017) | 20.781(±8.367) | 1.231(±0.014) | 11.823(±4.643) |
| | Deep Coral | SB | 0.014(±0.000) | 0.511(±0.420) | 0.548(±0.031) | 8.679(±5.518) | 0.018(±0.000) | 1.597(±1.222) | 0.033(±0.000) | 3.385(±2.597) | 1.970(±0.010) | 41.196(±43.492) | 1.227(±0.015) | 22.041(±21.683) |
| | Deep Coral | TB | 0.014(±0.000) | 0.212(±0.012) | 0.590(±0.045) | 4.547(±0.361) | 0.018(±0.000) | 0.679(±0.050) | 0.033(±0.001) | 1.427(±0.095) | 1.992(±0.028) | 13.829(±0.609) | 1.237(±0.018) | 8.097(±0.242) |
| | - | - | 0.023(±0.001) | 0.469(±0.055) | 2.240(±0.001) | 11.474(±0.290) | 0.026(±0.001) | 1.225(±0.165) | 0.051(±0.002) | 2.519(±0.385) | 2.860(±0.138) | 27.611(±5.693) | 1.674(±0.071) | 16.226(±3.967) |
| PointNet | DANN | DEV | 0.020(±0.001) | 1.093(±0.052) | 2.238(±0.002) | 17.985(±0.472) | 0.027(±0.002) | 3.313(±0.129) | 0.053(±0.004) | 7.055(±0.289) | 2.954(±0.179) | 101.784(±7.299) | 1.714(±0.101) | 51.655(±3.392) |
| | DANN | IWV | 0.020(±0.001) | 0.974(±0.419) | 2.243(±0.008) | 16.949(±3.600) | 0.028(±0.002) | 2.953(±1.232) | 0.054(±0.003) | 6.263(±2.630) | 2.949(±0.177) | 89.454(±42.987) | 1.727(±0.102) | 45.685(±21.204) |
| | DANN | SB | 0.019(±0.001) | 0.951(±0.347) | 2.339(±0.010) | 16.497(±3.351) | 0.027(±0.001) | 2.906(±1.015) | 0.052(±0.003) | 6.165(±2.173) | 2.886(±0.135) | 85.396(±36.953) | 1.679(±0.085) | 43.890(±17.807) |
| | DANN | TB | 0.020(±0.001) | 0.336(±0.054) | 2.239(±0.002) | 11.137(±0.329) | 0.028(±0.001) | 1.092(±0.206) | 0.052(±0.002) | 2.312(±0.421) | 2.988(±0.138) | 22.461(±4.074) | 1.733(±0.060) | 12.876(±2.085) |
| | CMD | DEV | 0.020(±0.001) | 1.030(±0.374) | 2.240(±0.002) | 17.213(±3.515) | 0.028(±0.001) | 3.196(±1.087) | 0.054(±0.003) | 6.777(±2.307) | 2.987(±0.188) | 89.470(±39.163) | 1.746(±0.085) | 45.786(±18.799) |
| | CMD | IWV | 0.019(±0.001) | 1.243(±0.047) | 2.240(±0.001) | 19.180(±0.409) | 0.028(±0.002) | 3.779(±0.111) | 0.054(±0.002) | 8.037(±0.257) | 2.996(±0.112) | 113.164(±5.966) | 1.758(±0.059) | 57.501(±3.749) |
| | CMD | SB | 0.019(±0.001) | 0.387(±0.059) | 2.241(±0.002) | 11.327(±0.574) | 0.027(±0.001) | 1.201(±0.297) | 0.051(±0.001) | 2.511(±0.699) | 2.852(±0.079) | 27.922(±6.676) | 1.675(±0.053) | 16.105(±3.828) |
| | CMD | TB | 0.019(±0.001) | 0.353(±0.078) | 2.240(±0.002) | 11.231(±0.508) | 0.026(±0.000) | 1.147(±0.284) | 0.051(±0.001) | 2.402(±0.634) | 2.843(±0.083) | 23.881(±4.994) | 1.684(±0.060) | 13.680(±2.721) |
| | Deep Coral | DEV | 0.020(±0.001) | 1.036(±0.102) | 2.241(±0.002) | 17.119(±1.256) | 0.028(±0.001) | 3.071(±0.374) | 0.055(±0.003) | 6.501(±0.887) | 3.003(±0.133) | 96.974(±10.676) | 1.768(±0.079) | 50.257(±3.638) |
| | Deep Coral | IWV | 0.020(±0.001) | 1.048(±0.167) | 2.238(±0.003) | 17.395(±1.880) | 0.028(±0.001) | 3.077(±0.508) | 0.055(±0.002) | 6.461(±1.120) | 2.984(±0.108) | 100.276(±20.956) | 1.775(±0.050) | 52.079(±8.798) |
| | Deep Coral | SB | 0.019(±0.000) | 0.977(±0.158) | 2.243(±0.002) | 16.764(±1.497) | 0.027(±0.001) | 2.947(±0.409) | 0.052(±0.001) | 6.257(±0.856) | 2.866(±0.081) | 88.933(±21.502) | 1.677(±0.043) | 45.919(±10.860) |
| | Deep Coral | TB | 0.019(±0.000) | 0.346(±0.078) | 2.239(±0.001) | 11.099(±0.287) | 0.027(±0.001) | 1.100(±0.270) | 0.051(±0.001) | 2.304(±0.618) | 2.857(±0.089) | 24.024(±6.005) | 1.693(±0.037) | 13.911(±3.132) |
| | - | - | 0.028(±0.001) | * | 0.580(±0.035) | * | 0.036(±0.001) | * | 0.070(±0.002) | * | 3.541(±0.094) | * | 2.056(±0.041) | * |
| Transolver | DANN | DEV | 0.024(±0.001) | 1.329(±0.053) | 0.572(±0.020) | 19.399(±0.646) | 0.038(±0.002) | 3.855(±0.131) | 0.075(±0.004) | 8.177(±0.272) | 3.562(±0.107) | 137.463(±8.757) | 2.072(±0.045) | 68.023(±3.389) |
| | DANN | IWV | 0.023(±0.001) | * | 0.563(±0.031) | * | 0.036(±0.002) | * | 0.072(±0.003) | * | 3.510(±0.112) | * | 2.052(±0.058) | * |
| | DANN | SB | 0.023(±0.000) | * | 0.557(±0.026) | * | 0.035(±0.001) | * | 0.069(±0.001) | * | 3.420(±0.039) | * | 1.989(±0.016) | * |
| | DANN | TB | 0.023(±0.001) | 1.248(±0.044) | 0.581(±0.052) | 18.346(±0.531) | 0.036(±0.002) | 3.634(±0.123) | 0.072(±0.004) | 7.700(±0.270) | 3.483(±0.129) | 126.739(±6.274) | 2.032(±0.073) | 63.423(±2.244) |
| | CMD | DEV | 0.024(±0.001) | 0.945(±0.385) | 0.609(±0.039) | 14.247(±5.368) | 0.037(±0.001) | 2.836(±1.064) | 0.074(±0.002) | 6.058(±2.217) | 3.586(±0.092) | 86.716(±48.985) | 2.071(±0.043) | 44.557(±22.613) |
| | CMD | IWV | 0.024(±0.001) | 2.630(±3.515) | 0.598(±0.040) | 78.553(±130.657) | 0.037(±0.002) | 4.817(±1.474) | 0.073(±0.003) | 12.720(±14.409) | 3.603(±0.105) | 350.914(±536.759) | 2.075(±0.056) | 251.012(±418.842) |
| | CMD | SB | 0.023(±0.001) | 0.899(±0.359) | 0.589(±0.026) | 13.868(±5.333) | 0.035(±0.001) | 2.743(±1.054) | 0.070(±0.003) | 5.800(±2.236) | 3.526(±0.128) | 77.730(±38.121) | 2.034(±0.051) | 41.268(±18.879) |
| | CMD | TB | 0.024(±0.001) | 0.567(±0.137) | 0.615(±0.047) | 9.350(±2.012) | 0.038(±0.002) | 1.798(±0.445) | 0.074(±0.003) | 3.834(±1.010) | 3.599(±0.156) | 41.982(±11.590) | 2.085(±0.092) | 23.189(±5.853) |
| | Deep Coral | DEV | 0.023(±0.001) | 3.231(±4.873) | 0.596(±0.014) | 91.582(±159.064) | 0.035(±0.002) | 9.198(±13.552) | 0.072(±0.003) | 22.590(±34.680) | 3.483(±0.170) | 253.494(±372.755) | 2.027(±0.084) | 174.785(±278.902) |
| | Deep Coral | IWV | 0.023(±0.001) | * | 0.606(±0.052) | * | 0.036(±0.003) | * | 0.072(±0.006) | * | 3.510(±0.124) | * | 2.035(±0.069) | * |
| | Deep Coral | SB | 0.023(±0.001) | 3.600(±4.655) | 0.583(±0.017) | 94.451(±157.174) | 0.034(±0.001) | 10.287(±12.902) | 0.068(±0.004) | 25.566(±32.972) | 3.409(±0.076) | 268.916(±363.263) | 1.989(±0.047) | 198.891(±265.635) |
| | Deep Coral | TB | 0.024(±0.001) | 0.656(±0.187) | 0.589(±0.020) | 10.200(±2.893) | 0.037(±0.001) | 1.985(±0.588) | 0.073(±0.003) | 4.247(±1.284) | 3.527(±0.051) | 53.775(±18.336) | 2.045(±0.046) | 29.340(±8.568) |

## A.2  Sheet Metal Forming

Table 4: Mean (± standard deviation) of RMSE across four seeds on the *sheet metal forming* dataset. Bold values indicate the best target domain performance across all normalized fields. Underlined entries mark the best performing UDA algorithm and unsupervised model selection strategy per model. Asterisks denote unstable models (error more than 10× higher than others).

| Model | DA Algorithm | Model Selection | All fields normalized Avg (-) SRC | TGT | Deformation (mm) SRC | TGT | Logarithmic strain (×10⁻²) SRC | TGT | Equivalent plastic strain (×10⁻²) SRC | TGT | Mises stress (MPa) SRC | TGT | Stress (MPa) SRC | TGT |
|---|---|---|---|---|---|---|---|---|---|---|---|---|---|---|
| | - | - | 0.070(±0.002) | 0.376(±0.028) | 1.411(±0.070) | 1.939(±0.530) | 0.024(±0.001) | 0.156(±0.014) | 0.043(±0.001) | 0.272(±0.026) | 11.022(±0.324) | 46.097(±4.911) | 5.548(±0.198) | 31.225(±1.554) |
| GraphSAGE | DANN | DEV | 0.056(±0.004) | * | 1.347(±0.045) | 16.199(±21.097) | 0.023(±0.001) | 0.965(±1.238) | 0.042(±0.003) | * | 10.597(±0.564) | 406.556(±403.135) | 5.334(±0.299) | 177.376(±187.164) |
| | DANN | IWV | 0.057(±0.003) | 0.329(±0.027) | 1.406(±0.071) | 2.095(±0.188) | 0.023(±0.001) | 0.158(±0.010) | 0.042(±0.003) | 0.269(±0.011) | 10.758(±0.277) | 52.401(±7.908) | 5.387(±0.134) | 34.644(±4.404) |
| | DANN | SB | 0.055(±0.002) | 1.139(±0.411) | 1.404(±0.055) | 7.810(±6.066) | 0.023(±0.001) | 0.467(±0.147) | 0.040(±0.001) | 0.921(±0.452) | 10.732(±0.406) | 186.098(±37.057) | 5.372(±0.168) | 94.370(±16.943) |
| | DANN | TB | 0.057(±0.003) | 0.323(±0.025) | 1.416(±0.055) | 2.021(±0.156) | 0.023(±0.001) | 0.156(±0.010) | 0.042(±0.003) | 0.265(±0.004) | 10.728(±0.218) | 49.234(±5.606) | 5.405(±0.167) | 33.375(±4.786) |
| | CMD | DEV | 0.055(±0.002) | 0.857(±0.475) | 1.355(±0.058) | 6.409(±3.478) | 0.022(±0.001) | 0.380(±0.177) | 0.041(±0.002) | 0.645(±0.290) | 10.590(±0.343) | 145.233(±79.730) | 5.287(±0.140) | 87.123(±42.179) |
| | CMD | IWV | 0.055(±0.001) | 0.407(±0.124) | 1.326(±0.031) | 2.455(±1.014) | 0.022(±0.000) | 0.201(±0.057) | 0.041(±0.000) | 0.336(±0.065) | 10.730(±0.065) | 61.665(±20.293) | 5.354(±0.115) | 37.004(±7.364) |
| | CMD | SB | 0.055(±0.001) | 0.569(±0.306) | 1.433(±0.024) | 4.708(±4.280) | 0.022(±0.000) | 0.290(±0.160) | 0.040(±0.000) | 0.497(±0.273) | 10.550(±0.163) | 99.069(±58.190) | 5.299(±0.071) | 55.134(±35.744) |
| | CMD | TB | 0.055(±0.001) | 0.289(±0.036) | 1.345(±0.059) | 2.028(±0.798) | 0.023(±0.000) | 0.139(±0.017) | 0.042(±0.001) | 0.243(±0.028) | 10.828(±0.169) | 43.746(±5.836) | 5.437(±0.110) | 29.606(±3.478) |
| | Deep Coral | DEV | 0.054(±0.001) | 0.411(±0.103) | 1.347(±0.048) | 3.347(±2.232) | 0.021(±0.001) | 0.185(±0.031) | 0.039(±0.001) | 0.330(±0.064) | 10.355(±0.455) | 65.861(±28.277) | 5.206(±0.190) | 43.062(±14.129) |
| | Deep Coral | IWV | 0.055(±0.002) | 0.353(±0.075) | 1.389(±0.055) | 2.449(±1.115) | 0.022(±0.001) | 0.170(±0.032) | 0.041(±0.002) | 0.304(±0.078) | 10.585(±0.289) | 48.326(±6.560) | 5.320(±0.174) | 34.667(±5.465) |
| | Deep Coral | SB | 0.056(±0.002) | 0.364(±0.105) | 1.392(±0.071) | 2.386(±0.735) | 0.022(±0.001) | 0.177(±0.055) | 0.041(±0.001) | 0.310(±0.090) | 10.744(±0.189) | 52.764(±11.554) | 5.368(±0.092) | 35.332(±8.182) |
| | Deep Coral | TB | 0.056(±0.003) | 0.287(±0.011) | 1.395(±0.068) | 1.825(±0.369) | 0.023(±0.001) | 0.137(±0.007) | 0.041(±0.002) | 0.242(±0.008) | 10.781(±0.433) | 44.161(±3.225) | 5.398(±0.179) | 29.228(±1.411) |
| | - | - | 0.077(±0.011) | 0.226(±0.047) | 2.012(±0.149) | 2.556(±0.948) | 0.024(±0.004) | 0.087(±0.022) | 0.045(±0.007) | 0.160(±0.039) | 11.357(±2.106) | 31.435(±6.317) | 8.067(±0.634) | 16.525(±3.262) |
| PointNet | DANN | DEV | 0.066(±0.003) | 1.195(±1.934) | 2.243(±0.041) | 6.185(±6.903) | 0.024(±0.001) | 0.709(±1.134) | 0.045(±0.003) | 1.528(±2.648) | 11.665(±0.483) | 129.318(±178.366) | 8.505(±0.083) | 101.783(±163.427) |
| | DANN | IWV | 0.067(±0.006) | 0.316(±0.171) | 2.283(±0.052) | 5.000(±4.861) | 0.025(±0.002) | 0.155(±0.081) | 0.047(±0.005) | 0.281(±0.149) | 12.151(±1.359) | 58.156(±38.050) | 8.631(±0.245) | 27.216(±16.145) |
| | DANN | SB | 0.067(±0.005) | 0.359(±0.153) | 2.250(±0.022) | 5.573(±4.577) | 0.025(±0.002) | 0.181(±0.076) | 0.047(±0.004) | 0.328(±0.138) | 12.090(±1.146) | 63.622(±34.926) | 8.522(±0.221) | 30.676(±14.620) |
| | DANN | TB | 0.076(±0.004) | 0.166(±0.008) | 2.270(±0.037) | 2.089(±0.144) | 0.028(±0.001) | 0.084(±0.014) | 0.053(±0.002) | 0.149(±0.016) | 14.069(±1.203) | 24.299(±2.097) | 9.041(±0.253) | 13.427(±0.788) |
| | CMD | DEV | 0.089(±0.037) | 0.329(±0.141) | 2.414(±0.373) | 4.199(±2.432) | 0.038(±0.024) | 0.162(±0.069) | 0.071(±0.045) | 0.280(±0.111) | 14.104(±3.213) | 61.546(±35.760) | 9.417(±1.408) | 28.416(±13.163) |
| | CMD | IWV | 0.071(±0.002) | 0.242(±0.148) | 2.263(±0.056) | 2.685(±0.972) | 0.026(±0.002) | 0.117(±0.071) | 0.050(±0.002) | 0.213(±0.126) | 12.925(±0.692) | 46.808(±38.805) | 8.806(±0.188) | 20.683(±12.572) |
| | CMD | SB | 0.060(±0.006) | 0.252(±0.066) | 1.988(±0.069) | 3.698(±1.484) | 0.022(±0.002) | 0.124(±0.029) | 0.042(±0.005) | 0.221(±0.049) | 10.166(±1.459) | 38.406(±13.599) | 7.737(±0.316) | 20.153(±5.512) |
| | CMD | TB | 0.069(±0.006) | 0.173(±0.013) | 2.099(±0.124) | 2.114(±0.141) | 0.026(±0.003) | 0.089(±0.011) | 0.049(±0.005) | 0.158(±0.019) | 12.260(±0.750) | 25.184(±1.660) | 8.365(±0.388) | 13.693(±0.839) |
| | Deep Coral | DEV | 0.067(±0.008) | 0.228(±0.065) | 2.201(±0.189) | 2.613(±0.839) | 0.025(±0.003) | 0.119(±0.040) | 0.046(±0.006) | 0.213(±0.067) | 12.087(±1.995) | 36.983(±12.354) | 8.439(±0.665) | 18.516(±5.099) |
| | Deep Coral | IWV | 0.064(±0.006) | 0.190(±0.027) | 2.196(±0.185) | 2.324(±0.411) | 0.024(±0.002) | 0.092(±0.013) | 0.044(±0.005) | 0.166(±0.022) | 11.283(±1.392) | 32.908(±5.779) | 8.302(±0.562) | 16.048(±2.999) |
| | Deep Coral | SB | 0.060(±0.009) | 0.182(±0.021) | 2.042(±0.185) | 2.555(±0.422) | 0.022(±0.004) | 0.084(±0.011) | 0.042(±0.008) | 0.150(±0.023) | 10.156(±2.001) | 31.345(±5.362) | 7.837(±0.674) | 16.017(±2.153) |
| | Deep Coral | TB | 0.069(±0.014) | **0.156(±0.006)** | 2.129(±0.184) | 2.004(±0.051) | 0.026(±0.006) | 0.085(±0.006) | 0.049(±0.011) | 0.140(±0.009) | 12.320(±3.129) | 22.942(±1.429) | 8.432(±0.932) | 12.967(±0.350) |
| | - | - | 0.070(±0.002) | 0.166(±0.029) | 1.168(±0.012) | 1.189(±0.293) | 0.022(±0.001) | 0.070(±0.015) | 0.041(±0.001) | 0.126(±0.029) | 12.862(±0.461) | 23.014(±4.849) | 6.033(±0.161) | 10.852(±1.952) |
| Transolver | DANN | DEV | 0.057(±0.002) | 0.206(±0.051) | 1.211(±0.062) | 2.625(±1.493) | 0.021(±0.001) | 0.103(±0.022) | 0.040(±0.001) | 0.187(±0.038) | 12.275(±0.537) | 36.777(±15.101) | 5.787(±0.203) | 17.571(±6.801) |
| | DANN | IWV | 0.056(±0.003) | 0.165(±0.026) | 1.194(±0.049) | 1.473(±0.537) | 0.021(±0.001) | 0.081(±0.011) | 0.040(±0.002) | 0.150(±0.023) | 12.223(±0.559) | 26.736(±6.986) | 5.764(±0.277) | 13.037(±3.317) |
| | DANN | SB | 0.057(±0.002) | 0.172(±0.016) | 1.207(±0.062) | 1.679(±0.366) | 0.021(±0.001) | 0.085(±0.006) | 0.040(±0.002) | 0.157(±0.012) | 12.074(±0.284) | 28.661(±5.284) | 5.709(±0.179) | 13.862(±2.528) |
| | DANN | TB | 0.056(±0.002) | 0.133(±0.016) | 1.249(±0.054) | 1.205(±0.276) | 0.021(±0.001) | 0.064(±0.013) | 0.041(±0.001) | 0.117(±0.025) | 12.560(±0.653) | 21.245(±1.910) | 5.924(±0.299) | 10.337(±0.834) |
| | CMD | DEV | 0.058(±0.002) | 0.286(±0.118) | 1.233(±0.062) | 4.088(±3.003) | 0.022(±0.001) | 0.142(±0.058) | 0.042(±0.001) | 0.255(±0.104) | 12.696(±0.924) | 51.628(±29.111) | 5.958(±0.363) | 26.089(±13.258) |
| | CMD | IWV | 0.056(±0.002) | 0.209(±0.096) | 1.200(±0.051) | 2.431(±1.533) | 0.021(±0.001) | 0.108(±0.054) | 0.040(±0.001) | 0.192(±0.092) | 12.080(±0.396) | 31.566(±13.954) | 5.712(±0.172) | 17.061(±8.722) |
| | CMD | SB | 0.056(±0.002) | 0.235(±0.097) | 1.214(±0.063) | 2.739(±1.545) | 0.021(±0.001) | 0.122(±0.053) | 0.040(±0.001) | 0.215(±0.090) | 12.145(±0.515) | 35.915(±14.900) | 5.731(±0.224) | 19.679(±9.245) |
| | CMD | TB | 0.062(±0.001) | 0.131(±0.008) | 1.249(±0.023) | 1.023(±0.023) | 0.023(±0.000) | 0.065(±0.005) | 0.044(±0.001) | 0.117(±0.007) | 13.505(±0.428) | 20.285(±1.747) | 6.326(±0.169) | 9.821(±0.838) |
| | Deep Coral | DEV | 0.058(±0.001) | **0.159(±0.011)** | 1.230(±0.033) | 1.386(±0.287) | 0.022(±0.000) | 0.081(±0.006) | 0.041(±0.001) | 0.146(±0.009) | 12.885(±0.257) | 25.049(±2.398) | 6.026(±0.065) | 12.572(±1.158) |
| | Deep Coral | IWV | 0.057(±0.001) | 0.261(±0.203) | 1.206(±0.008) | 3.011(±3.099) | 0.021(±0.001) | 0.133(±0.107) | 0.041(±0.001) | 0.240(±0.192) | 12.595(±0.275) | 44.262(±37.731) | 5.921(±0.116) | 22.722(±19.867) |
| | Deep Coral | SB | 0.057(±0.001) | 0.263(±0.201) | 1.199(±0.019) | 3.277(±2.944) | 0.021(±0.001) | 0.135(±0.106) | 0.040(±0.001) | 0.244(±0.189) | 12.509(±0.180) | 44.318(±37.691) | 5.878(±0.082) | 22.645(±19.921) |
| | Deep Coral | TB | 0.059(±0.001) | 0.138(±0.014) | 1.227(±0.016) | 0.957(±0.036) | 0.022(±0.000) | 0.068(±0.012) | 0.042(±0.001) | 0.124(±0.023) | 12.970(±0.502) | 22.062(±2.213) | 6.080(±0.207) | 10.846(±0.704) |

## A.3 Electric Motor Design

Table 5: Mean ($\pm$ standard deviation) of RMSE across four seeds on the *electric motor design* dataset. Bold values indicate the best target domain performance across all normalized fields. Underlined entries mark the best performing UDA algorithm and unsupervised model selection strategy per model. Asterisks denote unstable models (error more than 10$\times$ higher than others).

| Model | DA Algorithm | Model Selection | All fields normalized avg (-) | | Deformation (m) | | Logarithmic strain ($\times 10^{-2}$) | | Principal strain ($\times 10^{-2}$) | | Stress (MPa) | | Cauchy stress (MPa) | | Mises stress (MPa) | | Principal stress (MPa) | | Total strain ($\times 10^{-2}$) | |
|---|---|---|---|---|---|---|---|---|---|---|---|---|---|---|---|---|---|---|---|---|
| | | | SRC | TGT | SRC | TGT | SRC | TGT | SRC | TGT | SRC | TGT | SRC | TGT | SRC | TGT | SRC | TGT | SRC | TGT |
| GraphSAGE | - | - | 0.317($\pm$0.004) | 0.375($\pm$0.006) | 0.002($\pm$0.001) | 0.001($\pm$0.000) | 0.008($\pm$0.000) | 0.010($\pm$0.000) | 0.008($\pm$0.000) | 0.009($\pm$0.000) | 10.786($\pm$0.179) | 12.768($\pm$0.213) | 10.806($\pm$0.180) | 12.796($\pm$0.214) | 24.128($\pm$0.736) | 29.458($\pm$0.882) | 12.317($\pm$0.252) | 14.578($\pm$0.318) | 0.007($\pm$0.000) | 0.009($\pm$0.000) |
| | DANN | DEV | 0.314($\pm$0.023) | 0.443($\pm$0.071) | 0.003($\pm$0.000) | 0.002($\pm$0.000) | 0.009($\pm$0.001) | 0.012($\pm$0.002) | 0.008($\pm$0.001) | 0.012($\pm$0.002) | 11.443($\pm$0.760) | 16.354($\pm$2.726) | 11.463($\pm$0.761) | 16.387($\pm$2.730) | 25.734($\pm$1.822) | 38.735($\pm$6.880) | 13.059($\pm$0.826) | 18.779($\pm$3.137) | 0.008($\pm$0.001) | 0.011($\pm$0.002) |
| | DANN | IWV | 0.291($\pm$0.002) | 0.346($\pm$0.006) | 0.002($\pm$0.000) | 0.002($\pm$0.001) | 0.008($\pm$0.000) | 0.009($\pm$0.000) | 0.008($\pm$0.000) | 0.009($\pm$0.000) | 10.695($\pm$0.052) | 12.713($\pm$0.241) | 10.715($\pm$0.052) | 12.740($\pm$0.242) | 24.033($\pm$0.146) | 29.295($\pm$0.897) | 12.243($\pm$0.056) | 14.529($\pm$0.328) | 0.007($\pm$0.000) | 0.009($\pm$0.000) |
| | DANN | SB | 0.293($\pm$0.001) | 0.347($\pm$0.006) | 0.003($\pm$0.001) | 0.002($\pm$0.001) | 0.008($\pm$0.000) | 0.010($\pm$0.000) | 0.008($\pm$0.000) | 0.009($\pm$0.000) | 10.760($\pm$0.025) | 12.728($\pm$0.220) | 10.779($\pm$0.025) | 12.755($\pm$0.221) | 24.195($\pm$0.163) | 29.318($\pm$0.913) | 12.336($\pm$0.066) | 14.557($\pm$0.317) | 0.007($\pm$0.000) | 0.009($\pm$0.000) |
| | DANN | TB | 0.297($\pm$0.004) | 0.343($\pm$0.006) | 0.002($\pm$0.000) | 0.002($\pm$0.000) | 0.008($\pm$0.000) | 0.009($\pm$0.000) | 0.008($\pm$0.000) | 0.009($\pm$0.000) | 10.932($\pm$0.141) | 12.608($\pm$0.246) | 10.952($\pm$0.141) | 12.635($\pm$0.247) | 24.660($\pm$0.404) | 29.001($\pm$0.820) | 12.517($\pm$0.177) | 14.431($\pm$0.322) | 0.007($\pm$0.000) | 0.009($\pm$0.000) |
| | CMD | DEV | 0.299($\pm$0.019) | 0.395($\pm$0.052) | 0.002($\pm$0.000) | 0.002($\pm$0.000) | 0.008($\pm$0.000) | 0.011($\pm$0.002) | 0.008($\pm$0.000) | 0.011($\pm$0.002) | 10.919($\pm$0.624) | 14.550($\pm$2.044) | 10.939($\pm$0.624) | 14.580($\pm$2.048) | 24.480($\pm$1.528) | 34.337($\pm$6.047) | 12.483($\pm$0.694) | 16.672($\pm$2.506) | 0.007($\pm$0.000) | 0.010($\pm$0.001) |
| | CMD | IWV | 0.294($\pm$0.004) | 0.379($\pm$0.060) | 0.002($\pm$0.000) | 0.002($\pm$0.001) | 0.008($\pm$0.000) | 0.010($\pm$0.002) | 0.008($\pm$0.000) | 0.010($\pm$0.002) | 10.802($\pm$0.151) | 13.990($\pm$2.340) | 10.822($\pm$0.151) | 14.020($\pm$2.344) | 24.354($\pm$0.539) | 32.782($\pm$6.804) | 12.380($\pm$0.180) | 16.014($\pm$2.833) | 0.007($\pm$0.000) | 0.010($\pm$0.002) |
| | CMD | SB | 0.293($\pm$0.010) | 0.344($\pm$0.005) | 0.002($\pm$0.000) | 0.001($\pm$0.000) | 0.008($\pm$0.000) | 0.009($\pm$0.000) | 0.008($\pm$0.000) | 0.009($\pm$0.000) | 10.727($\pm$0.303) | 12.605($\pm$0.221) | 10.746($\pm$0.302) | 12.632($\pm$0.222) | 23.942($\pm$0.597) | 28.918($\pm$0.552) | 12.248($\pm$0.278) | 14.379($\pm$0.226) | 0.007($\pm$0.000) | 0.009($\pm$0.000) |
| | CMD | TB | 0.295($\pm$0.009) | 0.340($\pm$0.005) | 0.002($\pm$0.001) | 0.002($\pm$0.001) | 0.008($\pm$0.000) | 0.009($\pm$0.000) | 0.008($\pm$0.000) | 0.009($\pm$0.000) | 10.785($\pm$0.258) | 12.484($\pm$0.187) | 10.804($\pm$0.258) | 12.510($\pm$0.188) | 24.024($\pm$0.739) | 28.500($\pm$0.660) | 12.295($\pm$0.285) | 14.224($\pm$0.262) | 0.007($\pm$0.000) | 0.008($\pm$0.000) |
| | Deep Coral | DEV | 0.296($\pm$0.012) | 0.351($\pm$0.017) | 0.002($\pm$0.001) | 0.002($\pm$0.001) | 0.008($\pm$0.000) | 0.010($\pm$0.001) | 0.008($\pm$0.000) | 0.009($\pm$0.001) | 10.883($\pm$0.424) | 12.886($\pm$0.632) | 10.903($\pm$0.425) | 12.915($\pm$0.634) | 24.393($\pm$1.011) | 29.931($\pm$1.710) | 12.421($\pm$0.439) | 14.759($\pm$0.737) | 0.007($\pm$0.000) | 0.009($\pm$0.000) |
| | Deep Coral | IWV | 0.296($\pm$0.011) | 0.349($\pm$0.016) | 0.002($\pm$0.001) | 0.002($\pm$0.001) | 0.008($\pm$0.000) | 0.010($\pm$0.000) | 0.008($\pm$0.000) | 0.009($\pm$0.001) | 10.862($\pm$0.405) | 12.782($\pm$0.586) | 10.882($\pm$0.406) | 12.810($\pm$0.588) | 24.261($\pm$1.070) | 29.194($\pm$1.648) | 12.374($\pm$0.441) | 14.543($\pm$0.697) | 0.007($\pm$0.000) | 0.009($\pm$0.000) |
| | Deep Coral | SB | 0.288($\pm$0.004) | 0.351($\pm$0.008) | 0.003($\pm$0.000) | 0.002($\pm$0.001) | 0.008($\pm$0.000) | 0.010($\pm$0.000) | 0.007($\pm$0.000) | 0.009($\pm$0.000) | 10.543($\pm$0.129) | 12.915($\pm$0.312) | 10.562($\pm$0.129) | 12.943($\pm$0.313) | 23.450($\pm$0.270) | 30.013($\pm$1.011) | 12.075($\pm$0.169) | 14.773($\pm$0.350) | 0.007($\pm$0.000) | 0.009($\pm$0.000) |
| | Deep Coral | TB | 0.290($\pm$0.003) | 0.338($\pm$0.003) | 0.002($\pm$0.001) | 0.002($\pm$0.001) | 0.008($\pm$0.000) | 0.009($\pm$0.000) | 0.008($\pm$0.000) | 0.009($\pm$0.000) | 10.678($\pm$0.134) | 12.401($\pm$0.118) | 10.698($\pm$0.134) | 12.428($\pm$0.119) | 23.983($\pm$0.330) | 28.527($\pm$0.407) | 12.206($\pm$0.130) | 14.163($\pm$0.148) | 0.007($\pm$0.000) | 0.008($\pm$0.000) |
| PointNet | - | - | 0.319($\pm$0.050) | 0.396($\pm$0.048) | 0.002($\pm$0.001) | 0.002($\pm$0.001) | 0.008($\pm$0.001) | 0.010($\pm$0.001) | 0.008($\pm$0.001) | 0.010($\pm$0.001) | 10.714($\pm$1.624) | 13.389($\pm$1.583) | 10.731($\pm$1.626) | 13.417($\pm$1.584) | 23.666($\pm$3.655) | 30.654($\pm$3.391) | 12.090($\pm$1.910) | 15.146($\pm$1.847) | 0.007($\pm$0.001) | 0.009($\pm$0.001) |
| | DANN | DEV | 0.289($\pm$0.050) | 0.505($\pm$0.032) | 0.001($\pm$0.000) | 0.001($\pm$0.000) | 0.008($\pm$0.001) | 0.014($\pm$0.001) | 0.007($\pm$0.001) | 0.014($\pm$0.001) | 10.489($\pm$1.784) | 18.840($\pm$1.166) | 10.506($\pm$1.787) | 18.875($\pm$1.167) | 23.009($\pm$3.887) | 44.796($\pm$2.664) | 11.815($\pm$2.054) | 21.509($\pm$1.376) | 0.007($\pm$0.001) | 0.013($\pm$0.001) |
| | DANN | IWV | 0.275($\pm$0.037) | 0.444($\pm$0.085) | 0.001($\pm$0.000) | 0.001($\pm$0.000) | 0.007($\pm$0.001) | 0.012($\pm$0.002) | 0.007($\pm$0.001) | 0.012($\pm$0.002) | 9.973($\pm$1.270) | 16.409($\pm$3.248) | 9.990($\pm$1.271) | 16.442($\pm$3.253) | 21.849($\pm$2.570) | 38.862($\pm$7.799) | 11.214($\pm$1.442) | 18.724($\pm$3.694) | 0.007($\pm$0.001) | 0.011($\pm$0.002) |
| | DANN | SB | 0.269($\pm$0.037) | 0.434($\pm$0.121) | 0.002($\pm$0.001) | 0.002($\pm$0.001) | 0.007($\pm$0.001) | 0.012($\pm$0.003) | 0.007($\pm$0.001) | 0.012($\pm$0.003) | 9.749($\pm$1.272) | 16.051($\pm$4.549) | 9.766($\pm$1.273) | 16.083($\pm$4.555) | 21.424($\pm$2.685) | 37.862($\pm$10.930) | 10.980($\pm$1.463) | 18.344($\pm$5.326) | 0.007($\pm$0.001) | 0.011($\pm$0.003) |
| | DANN | TB | 0.279($\pm$0.053) | 0.343($\pm$0.052) | 0.001($\pm$0.000) | 0.000($\pm$0.000) | 0.008($\pm$0.001) | 0.009($\pm$0.001) | 0.007($\pm$0.001) | 0.009($\pm$0.001) | 10.117($\pm$1.888) | 12.449($\pm$1.838) | 10.135($\pm$1.891) | 12.476($\pm$1.841) | 22.144($\pm$4.215) | 28.299($\pm$3.982) | 11.426($\pm$2.183) | 14.100($\pm$2.119) | 0.007($\pm$0.001) | 0.009($\pm$0.001) |
| | CMD | DEV | 0.321($\pm$0.107) | 0.380($\pm$0.074) | 0.002($\pm$0.001) | 0.001($\pm$0.001) | 0.009($\pm$0.003) | 0.011($\pm$0.002) | 0.007($\pm$0.002) | 0.010($\pm$0.002) | 11.854($\pm$4.237) | 13.976($\pm$2.812) | 11.873($\pm$4.243) | 14.006($\pm$2.817) | 23.042($\pm$5.163) | 32.349($\pm$6.906) | 11.970($\pm$2.765) | 15.852($\pm$3.293) | 0.008($\pm$0.003) | 0.010($\pm$0.002) |
| | CMD | IWV | 0.470($\pm$0.454) | 0.353($\pm$0.073) | 0.006($\pm$0.009) | 0.002($\pm$0.001) | 0.013($\pm$0.014) | 0.010($\pm$0.002) | 0.011($\pm$0.010) | 0.009($\pm$0.002) | 17.801($\pm$18.040) | 12.930($\pm$2.776) | 17.825($\pm$18.057) | 12.958($\pm$2.781) | 34.334($\pm$31.237) | 29.373($\pm$6.774) | 20.747($\pm$21.808) | 14.613($\pm$3.036) | 0.012($\pm$0.013) | 0.009($\pm$0.002) |
| | CMD | SB | 0.471($\pm$0.453) | 0.353($\pm$0.073) | 0.006($\pm$0.009) | 0.002($\pm$0.001) | 0.014($\pm$0.014) | 0.010($\pm$0.002) | 0.011($\pm$0.010) | 0.009($\pm$0.002) | 17.862($\pm$18.009) | 12.998($\pm$2.799) | 17.886($\pm$18.026) | 13.027($\pm$2.803) | 34.519($\pm$31.151) | 29.782($\pm$6.876) | 20.804($\pm$21.778) | 14.717($\pm$3.074) | 0.012($\pm$0.013) | 0.009($\pm$0.002) |
| | CMD | TB | 0.252($\pm$0.048) | 0.314($\pm$0.059) | 0.002($\pm$0.000) | 0.001($\pm$0.000) | 0.007($\pm$0.001) | 0.009($\pm$0.002) | 0.006($\pm$0.001) | 0.008($\pm$0.002) | 9.109($\pm$1.686) | 11.382($\pm$2.094) | 9.125($\pm$1.689) | 11.408($\pm$2.097) | 19.744($\pm$3.712) | 25.760($\pm$5.145) | 10.278($\pm$1.857) | 12.934($\pm$2.404) | 0.006($\pm$0.001) | 0.008($\pm$0.001) |
| | Deep Coral | DEV | 0.254($\pm$0.034) | 0.327($\pm$0.031) | 0.002($\pm$0.001) | 0.002($\pm$0.001) | 0.007($\pm$0.001) | 0.009($\pm$0.001) | 0.006($\pm$0.001) | 0.009($\pm$0.001) | 9.185($\pm$1.182) | 11.852($\pm$1.071) | 9.201($\pm$1.184) | 11.877($\pm$1.073) | 20.106($\pm$2.391) | 27.164($\pm$1.924) | 10.390($\pm$1.267) | 13.490($\pm$1.162) | 0.006($\pm$0.001) | 0.008($\pm$0.001) |
| | Deep Coral | IWV | 0.259($\pm$0.033) | 0.318($\pm$0.012) | 0.002($\pm$0.001) | 0.002($\pm$0.001) | 0.007($\pm$0.001) | 0.009($\pm$0.000) | 0.007($\pm$0.001) | 0.009($\pm$0.000) | 9.318($\pm$1.064) | 11.525($\pm$0.391) | 9.334($\pm$1.065) | 11.551($\pm$0.391) | 20.385($\pm$2.250) | 26.522($\pm$0.984) | 10.519($\pm$1.137) | 13.139($\pm$0.507) | 0.006($\pm$0.001) | 0.008($\pm$0.000) |
| | Deep Coral | SB | 0.255($\pm$0.035) | 0.313($\pm$0.017) | 0.002($\pm$0.001) | 0.002($\pm$0.001) | 0.007($\pm$0.001) | 0.009($\pm$0.000) | 0.006($\pm$0.001) | 0.008($\pm$0.000) | 9.189($\pm$1.130) | 11.357($\pm$0.564) | 9.205($\pm$1.131) | 11.382($\pm$0.565) | 20.156($\pm$2.363) | 26.229($\pm$1.375) | 10.371($\pm$1.224) | 12.941($\pm$0.727) | 0.006($\pm$0.001) | 0.008($\pm$0.000) |
| | Deep Coral | TB | 0.255($\pm$0.035) | 0.313($\pm$0.017) | 0.002($\pm$0.001) | 0.002($\pm$0.001) | 0.007($\pm$0.001) | 0.009($\pm$0.000) | 0.006($\pm$0.001) | 0.008($\pm$0.000) | 9.189($\pm$1.130) | 11.357($\pm$0.564) | 9.205($\pm$1.131) | 11.382($\pm$0.565) | 20.156($\pm$2.363) | 26.229($\pm$1.375) | 10.371($\pm$1.224) | 12.941($\pm$0.727) | 0.006($\pm$0.001) | 0.008($\pm$0.000) |
| Transolver | - | - | 0.104($\pm$0.011) | 0.121($\pm$0.007) | 0.002($\pm$0.001) | 0.001($\pm$0.000) | 0.002($\pm$0.000) | 0.003($\pm$0.000) | 0.002($\pm$0.000) | 0.003($\pm$0.000) | 3.448($\pm$0.377) | 3.975($\pm$0.220) | 3.454($\pm$0.377) | 3.982($\pm$0.220) | 6.999($\pm$0.576) | 8.328($\pm$0.473) | 4.243($\pm$0.570) | 4.811($\pm$0.231) | 0.002($\pm$0.000) | 0.003($\pm$0.000) |
| | DANN | DEV | 0.088($\pm$0.002) | 0.111($\pm$0.006) | 0.001($\pm$0.000) | 0.001($\pm$0.000) | 0.002($\pm$0.000) | 0.003($\pm$0.000) | 0.002($\pm$0.000) | 0.003($\pm$0.000) | 3.150($\pm$0.064) | 3.953($\pm$0.216) | 3.155($\pm$0.064) | 3.961($\pm$0.216) | 6.455($\pm$0.162) | 8.334($\pm$0.554) | 3.825($\pm$0.086) | 4.774($\pm$0.233) | 0.002($\pm$0.000) | 0.003($\pm$0.000) |
| | DANN | IWV | 0.087($\pm$0.001) | 0.111($\pm$0.006) | 0.001($\pm$0.000) | 0.001($\pm$0.000) | 0.002($\pm$0.000) | 0.003($\pm$0.000) | 0.002($\pm$0.000) | 0.003($\pm$0.000) | 3.102($\pm$0.037) | 3.926($\pm$0.234) | 3.108($\pm$0.037) | 3.934($\pm$0.234) | 6.390($\pm$0.077) | 8.280($\pm$0.589) | 3.768($\pm$0.040) | 4.747($\pm$0.255) | 0.002($\pm$0.000) | 0.003($\pm$0.000) |
| | DANN | SB | 0.085($\pm$0.002) | 0.109($\pm$0.007) | 0.001($\pm$0.000) | 0.001($\pm$0.000) | 0.002($\pm$0.000) | 0.003($\pm$0.000) | 0.002($\pm$0.000) | 0.003($\pm$0.000) | 3.029($\pm$0.070) | 3.869($\pm$0.280) | 3.034($\pm$0.070) | 3.877($\pm$0.281) | 6.210($\pm$0.140) | 8.174($\pm$0.673) | 3.680($\pm$0.083) | 4.672($\pm$0.308) | 0.002($\pm$0.000) | 0.003($\pm$0.000) |
| | DANN | TB | 0.085($\pm$0.002) | 0.104($\pm$0.003) | 0.001($\pm$0.000) | 0.001($\pm$0.000) | 0.002($\pm$0.000) | 0.003($\pm$0.000) | 0.002($\pm$0.000) | 0.003($\pm$0.000) | 3.030($\pm$0.088) | 3.710($\pm$0.091) | 3.035($\pm$0.088) | 3.718($\pm$0.091) | 6.248($\pm$0.208) | 7.788($\pm$0.187) | 3.689($\pm$0.102) | 4.502($\pm$0.102) | 0.002($\pm$0.000) | 0.002($\pm$0.000) |
| | CMD | DEV | 0.089($\pm$0.004) | 0.112($\pm$0.007) | 0.001($\pm$0.000) | 0.001($\pm$0.000) | 0.002($\pm$0.000) | 0.003($\pm$0.000) | 0.002($\pm$0.000) | 0.003($\pm$0.000) | 3.176($\pm$0.136) | 3.950($\pm$0.204) | 3.182($\pm$0.136) | 3.958($\pm$0.205) | 6.536($\pm$0.266) | 8.223($\pm$0.396) | 3.851($\pm$0.152) | 4.782($\pm$0.255) | 0.002($\pm$0.000) | 0.003($\pm$0.000) |
| | CMD | IWV | 0.088($\pm$0.004) | 0.107($\pm$0.007) | 0.001($\pm$0.000) | 0.001($\pm$0.000) | 0.002($\pm$0.000) | 0.003($\pm$0.000) | 0.002($\pm$0.000) | 0.003($\pm$0.000) | 3.140($\pm$0.140) | 3.802($\pm$0.233) | 3.146($\pm$0.140) | 3.809($\pm$0.234) | 6.488($\pm$0.324) | 7.971($\pm$0.511) | 3.803($\pm$0.163) | 4.594($\pm$0.252) | 0.002($\pm$0.000) | 0.002($\pm$0.000) |
| | CMD | SB | 0.086($\pm$0.001) | 0.106($\pm$0.004) | 0.001($\pm$0.000) | 0.001($\pm$0.000) | 0.002($\pm$0.000) | 0.003($\pm$0.000) | 0.002($\pm$0.000) | 0.003($\pm$0.000) | 3.085($\pm$0.032) | 3.757($\pm$0.137) | 3.090($\pm$0.032) | 3.765($\pm$0.137) | 6.334($\pm$0.080) | 7.864($\pm$0.363) | 3.752($\pm$0.039) | 4.553($\pm$0.174) | 0.002($\pm$0.000) | 0.002($\pm$0.000) |
| | CMD | TB | 0.086($\pm$0.002) | 0.103($\pm$0.003) | 0.001($\pm$0.000) | 0.001($\pm$0.000) | 0.002($\pm$0.000) | 0.003($\pm$0.000) | 0.002($\pm$0.000) | 0.003($\pm$0.000) | 3.061($\pm$0.081) | 3.670($\pm$0.115) | 3.066($\pm$0.081) | 3.677($\pm$0.116) | 6.294($\pm$0.183) | 7.646($\pm$0.237) | 3.720($\pm$0.097) | 4.447($\pm$0.125) | 0.002($\pm$0.000) | 0.002($\pm$0.000) |
| | Deep Coral | DEV | 0.087($\pm$0.002) | 0.105($\pm$0.002) | 0.001($\pm$0.000) | 0.001($\pm$0.000) | 0.002($\pm$0.000) | 0.003($\pm$0.000) | 0.002($\pm$0.000) | 0.003($\pm$0.000) | 3.093($\pm$0.076) | 3.720($\pm$0.076) | 3.099($\pm$0.076) | 3.727($\pm$0.076) | 6.394($\pm$0.171) | 7.825($\pm$0.181) | 3.768($\pm$0.095) | 4.533($\pm$0.102) | 0.002($\pm$0.000) | 0.002($\pm$0.000) |
| | Deep Coral | IWV | 0.161($\pm$0.146) | 0.104($\pm$0.003) | 0.002($\pm$0.002) | 0.001($\pm$0.000) | 0.003($\pm$0.002) | 0.003($\pm$0.000) | 0.004($\pm$0.003) | 0.003($\pm$0.000) | 4.747($\pm$3.245) | 3.711($\pm$0.111) | 4.751($\pm$3.242) | 3.718($\pm$0.111) | 11.535($\pm$10.150) | 7.784($\pm$0.285) | 6.294($\pm$4.970) | 4.524($\pm$0.146) | 0.003($\pm$0.001) | 0.002($\pm$0.000) |
| | Deep Coral | SB | 0.085($\pm$0.003) | **0.103($\pm$0.004)** | 0.001($\pm$0.000) | 0.001($\pm$0.000) | 0.002($\pm$0.000) | 0.003($\pm$0.000) | 0.002($\pm$0.000) | 0.003($\pm$0.000) | 3.042($\pm$0.100) | 3.658($\pm$0.142) | 3.048($\pm$0.100) | 3.664($\pm$0.142) | 6.261($\pm$0.214) | 7.678($\pm$0.339) | 3.695($\pm$0.117) | 4.446($\pm$0.162) | 0.002($\pm$0.000) | 0.002($\pm$0.000) |
| | Deep Coral | TB | 0.086($\pm$0.003) | 0.102($\pm$0.003) | 0.001($\pm$0.000) | 0.001($\pm$0.000) | 0.002($\pm$0.000) | 0.003($\pm$0.000) | 0.002($\pm$0.000) | 0.003($\pm$0.000) | 3.067($\pm$0.108) | 3.638($\pm$0.123) | 3.072($\pm$0.109) | 3.645($\pm$0.124) | 6.330($\pm$0.252) | 7.594($\pm$0.268) | 3.724($\pm$0.128) | 4.417($\pm$0.139) | 0.002($\pm$0.000) | 0.002($\pm$0.000) |

## A.4 Heatsink Design

Table 6: Mean ($\pm$ standard deviation) of RMSE across four seeds on the *heatsink design* dataset. Bold values indicate the best target domain performance across all normalized fields. Underlined entries mark the best performing UDA algorithm and unsupervised model selection strategy per model.

| Model | DA Algorithm | Model Selection | All fields normalized avg (-) | | Temperature (K) | | Velocity (m/s) | | Pressure (kPa) | |
|---|---|---|---|---|---|---|---|---|---|---|
| | | | SRC | TGT | SRC | TGT | SRC | TGT | SRC | TGT |
| PointNet | - | - | 0.525($\pm$0.026) | 0.568($\pm$0.030) | 15.581($\pm$1.535) | 21.126($\pm$2.365) | 0.054($\pm$0.002) | 0.044($\pm$0.000) | 0.386($\pm$0.034) | 1.879($\pm$0.239) |
| | DANN | DEV | 0.339($\pm$0.104) | 0.442($\pm$0.050) | 12.078($\pm$4.555) | 19.408($\pm$3.391) | 0.043($\pm$0.009) | 0.047($\pm$0.007) | 0.815($\pm$1.032) | 1.998($\pm$0.360) |
| | DANN | IWV | 0.289($\pm$0.056) | 0.429($\pm$0.052) | 10.167($\pm$2.894) | 18.172($\pm$3.222) | 0.040($\pm$0.008) | 0.047($\pm$0.007) | 0.283($\pm$0.071) | 1.806($\pm$0.145) |
| | DANN | SB | 0.228($\pm$0.016) | 0.494($\pm$0.026) | 6.668($\pm$1.013) | 20.129($\pm$2.380) | 0.031($\pm$0.002) | 0.055($\pm$0.002) | 0.207($\pm$0.014) | 2.103($\pm$0.615) |
| | DANN | TB | 0.304($\pm$0.036) | 0.397($\pm$0.019) | 10.964($\pm$1.411) | 15.719($\pm$1.387) | 0.041($\pm$0.005) | 0.043($\pm$0.002) | 0.331($\pm$0.141) | 1.908($\pm$0.232) |
| | CMD | DEV | 0.423($\pm$0.003) | 0.442($\pm$0.004) | 16.324($\pm$0.135) | 20.548($\pm$0.035) | 0.042($\pm$0.001) | 0.042($\pm$0.000) | 2.386($\pm$0.018) | 2.466($\pm$0.042) |
| | CMD | IWV | 0.239($\pm$0.008) | 0.480($\pm$0.020) | 7.577($\pm$0.479) | 18.524($\pm$1.213) | 0.033($\pm$0.001) | 0.051($\pm$0.002) | 0.193($\pm$0.005) | 2.455($\pm$0.118) |
| | CMD | SB | 0.238($\pm$0.007) | 0.475($\pm$0.025) | 7.433($\pm$0.330) | 18.460($\pm$1.300) | 0.033($\pm$0.001) | 0.051($\pm$0.002) | 0.199($\pm$0.009) | 2.373($\pm$0.157) |
| | CMD | TB | 0.302($\pm$0.086) | 0.442($\pm$0.018) | 10.801($\pm$4.087) | 17.800($\pm$2.256) | 0.037($\pm$0.004) | 0.046($\pm$0.004) | 0.757($\pm$1.077) | 2.289($\pm$0.108) |
| | Deep Coral | DEV | 0.275($\pm$0.071) | 0.394($\pm$0.048) | 9.324($\pm$3.565) | 18.021($\pm$2.349) | 0.038($\pm$0.010) | 0.044($\pm$0.006) | 0.239($\pm$0.084) | 0.988($\pm$0.479) |
| | Deep Coral | IWV | 0.275($\pm$0.071) | 0.394($\pm$0.048) | 9.324($\pm$3.565) | 18.021($\pm$2.349) | 0.038($\pm$0.010) | 0.044($\pm$0.006) | 0.239($\pm$0.084) | 0.988($\pm$0.479) |
| | Deep Coral | SB | 0.270($\pm$0.061) | 0.394($\pm$0.048) | 9.071($\pm$3.069) | 17.428($\pm$1.939) | 0.037($\pm$0.009) | 0.044($\pm$0.006) | 0.224($\pm$0.055) | 1.037($\pm$0.574) |
| | Deep Coral | TB | 0.343($\pm$0.063) | 0.384($\pm$0.042) | 12.763($\pm$3.067) | 18.517($\pm$2.502) | 0.047($\pm$0.009) | 0.042($\pm$0.004) | 0.324($\pm$0.103) | 1.439($\pm$0.427) |
| Transolver | - | - | 0.348($\pm$0.009) | 0.487($\pm$0.009) | 8.553($\pm$0.526) | 13.432($\pm$0.486) | 0.033($\pm$0.001) | 0.040($\pm$0.000) | 0.519($\pm$0.047) | 1.655($\pm$0.176) |
| | DANN | DEV | 0.275($\pm$0.042) | 0.433($\pm$0.030) | 9.629($\pm$2.784) | 17.110($\pm$1.633) | 0.035($\pm$0.006) | 0.048($\pm$0.004) | 0.486($\pm$0.043) | 1.871($\pm$0.135) |
| | DANN | IWV | 0.276($\pm$0.039) | 0.448($\pm$0.022) | 9.251($\pm$1.988) | 17.483($\pm$1.168) | 0.035($\pm$0.005) | 0.050($\pm$0.003) | 0.547($\pm$0.146) | 1.993($\pm$0.179) |
| | DANN | SB | 0.251($\pm$0.005) | 0.445($\pm$0.014) | 7.823($\pm$0.056) | 16.603($\pm$1.047) | 0.032($\pm$0.001) | 0.049($\pm$0.002) | 0.487($\pm$0.040) | 2.079($\pm$0.134) |
| | DANN | TB | 0.296($\pm$0.046) | 0.425($\pm$0.024) | 10.624($\pm$2.804) | 16.740($\pm$0.747) | 0.038($\pm$0.006) | 0.047($\pm$0.003) | 0.583($\pm$0.121) | 1.921($\pm$0.163) |
| | CMD | DEV | 0.412($\pm$0.006) | 0.495($\pm$0.014) | 16.426($\pm$0.267) | 22.584($\pm$0.912) | 0.038($\pm$0.001) | 0.047($\pm$0.001) | 2.509($\pm$0.119) | 2.926($\pm$0.150) |
| | CMD | IWV | 0.256($\pm$0.005) | 0.411($\pm$0.028) | 8.321($\pm$0.303) | 15.435($\pm$2.032) | 0.033($\pm$0.000) | 0.046($\pm$0.004) | 0.465($\pm$0.066) | 1.870($\pm$0.057) |
| | CMD | SB | 0.255($\pm$0.006) | 0.420($\pm$0.038) | 8.341($\pm$0.280) | 15.821($\pm$2.496) | 0.032($\pm$0.001) | 0.046($\pm$0.005) | 0.471($\pm$0.058) | 1.915($\pm$0.061) |
| | CMD | TB | 0.256($\pm$0.005) | 0.408($\pm$0.024) | 8.269($\pm$0.208) | 15.028($\pm$1.653) | 0.033($\pm$0.001) | 0.045($\pm$0.003) | 0.431($\pm$0.059) | 1.900($\pm$0.107) |
| | Deep Coral | DEV | 0.261($\pm$0.004) | 0.374($\pm$0.005) | 8.652($\pm$0.241) | 13.539($\pm$0.543) | 0.033($\pm$0.000) | 0.041($\pm$0.001) | 0.515($\pm$0.047) | 1.726($\pm$0.104) |
| | Deep Coral | IWV | 0.257($\pm$0.014) | 0.368($\pm$0.009) | 8.349($\pm$0.855) | 13.434($\pm$0.870) | 0.033($\pm$0.001) | 0.041($\pm$0.001) | 0.481($\pm$0.074) | 1.559($\pm$0.127) |
| | Deep Coral | SB | 0.245($\pm$0.005) | 0.372($\pm$0.015) | 7.783($\pm$0.388) | 13.367($\pm$0.909) | 0.032($\pm$0.001) | 0.041($\pm$0.002) | 0.388($\pm$0.014) | 1.719($\pm$0.188) |
| | Deep Coral | TB | 0.259($\pm$0.013) | 0.351($\pm$0.023) | 8.389($\pm$0.613) | 12.756($\pm$1.125) | 0.033($\pm$0.001) | 0.039($\pm$0.002) | 0.529($\pm$0.113) | 1.464($\pm$0.180) |
| UPT | - | - | 0.244($\pm$0.002) | 0.441($\pm$0.024) | 4.316($\pm$0.028) | 13.033($\pm$1.059) | 0.025($\pm$0.000) | 0.040($\pm$0.002) | 0.232($\pm$0.014) | 0.816($\pm$0.049) |
| | DANN | DEV | 0.188($\pm$0.011) | 0.446($\pm$0.026) | 4.651($\pm$0.781) | 15.580($\pm$0.609) | 0.026($\pm$0.002) | 0.050($\pm$0.003) | 0.223($\pm$0.013) | 2.165($\pm$0.302) |
| | DANN | IWV | 0.222($\pm$0.053) | 0.443($\pm$0.070) | 6.731($\pm$3.132) | 15.179($\pm$1.591) | 0.030($\pm$0.007) | 0.048($\pm$0.006) | 0.247($\pm$0.033) | 2.380($\pm$0.727) |
| | DANN | SB | 0.184($\pm$0.002) | 0.480($\pm$0.018) | 4.285($\pm$0.072) | 15.689($\pm$0.806) | 0.025($\pm$0.000) | 0.051($\pm$0.001) | 0.244($\pm$0.024) | 2.729($\pm$0.517) |
| | DANN | TB | 0.273($\pm$0.092) | 0.398($\pm$0.038) | 9.411($\pm$4.841) | 15.644($\pm$3.334) | 0.037($\pm$0.012) | 0.043($\pm$0.004) | 0.285($\pm$0.073) | 1.872($\pm$0.366) |
| | CMD | DEV | 0.210($\pm$0.055) | 0.406($\pm$0.046) | 5.994($\pm$3.353) | 14.289($\pm$2.054) | 0.028($\pm$0.007) | 0.046($\pm$0.005) | 0.236($\pm$0.022) | 1.874($\pm$0.394) |
| | CMD | IWV | 0.182($\pm$0.000) | 0.363($\pm$0.015) | 4.297($\pm$0.038) | 12.908($\pm$0.487) | 0.025($\pm$0.000) | 0.043($\pm$0.001) | 0.221($\pm$0.009) | 1.365($\pm$0.257) |
| | CMD | SB | 0.179($\pm$0.001) | 0.444($\pm$0.010) | 4.135($\pm$0.026) | 16.130($\pm$0.627) | 0.024($\pm$0.000) | 0.050($\pm$0.001) | 0.231($\pm$0.008) | 1.919($\pm$0.052) |
| | CMD | TB | 0.182($\pm$0.000) | 0.363($\pm$0.015) | 4.297($\pm$0.038) | 12.908($\pm$0.487) | 0.025($\pm$0.000) | 0.043($\pm$0.001) | 0.221($\pm$0.009) | 1.365($\pm$0.257) |
| | Deep Coral | DEV | 0.183($\pm$0.001) | 0.345($\pm$0.013) | 4.318($\pm$0.067) | 13.290($\pm$0.655) | 0.025($\pm$0.000) | 0.041($\pm$0.001) | 0.221($\pm$0.008) | 0.810($\pm$0.099) |
| | Deep Coral | IWV | 0.183($\pm$0.001) | 0.339($\pm$0.020) | 4.344($\pm$0.055) | 13.037($\pm$1.027) | 0.025($\pm$0.000) | 0.041($\pm$0.002) | 0.223($\pm$0.007) | 0.778($\pm$0.065) |
| | Deep Coral | SB | 0.182($\pm$0.000) | **0.325($\pm$0.008)** | 4.307($\pm$0.042) | 12.414($\pm$1.209) | 0.025($\pm$0.000) | 0.039($\pm$0.001) | 0.214($\pm$0.007) | 0.840($\pm$0.184) |
| | Deep Coral | TB | 0.182($\pm$0.000) | 0.321($\pm$0.008) | 4.347($\pm$0.039) | 12.637($\pm$0.949) | 0.025($\pm$0.000) | 0.039($\pm$0.001) | 0.218($\pm$0.012) | 0.792($\pm$0.122) |

# B  Distribution Shifts

Table 7 provides an overview of the parameter ranges chosen to define source and target domains for different task difficulties across all datasets. To gain more insights into the parameter importance besides the domain experts' opinion, we visualize the latent space of the conditioning network for all presented datasets in Figures 7 to 10.

Table 7: Defined distribution shifts (source and target domains) of each dataset and each difficulty.

| Dataset | Parameter | Difficulty | Source range (no. samples) | Target range (no. samples) |
|---|---|---|---|---|
| Rolling | Reduction $r$ $(-)$ | easy | [0.01, 0.13) (4000) | [0.13, 0.15] (750) |
| | | medium | [0.01, 0.115) (3500) | [0.115, 0.15] (1250) |
| | | hard | [0.01, 0.10) (3000) | [0.10, 0.15] (1750) |
| Forming | Thickness $t$ $(mm)$ | easy | [2, 4.8) (3060) | [4.8, 5] (255) |
| | | medium | [2, 4.3) (2550) | [4.3, 5] (765) |
| | | hard | [2, 4.1) (2295) | [4.1, 5] (1020) |
| Electric Motor | Rotor slot diameter 3 $d_{r3}$ $(mm)$ | easy | [100, 122)(2693) | [122, 126](504) |
| | | medium | [99, 120) (2143) | [120, 126] (1054) |
| | | hard | [99, 118) (1728) | [118, 126] (1469) |
| Heatsink | # fins | easy | [5, 13) (404) | [13, 14] (56) |
| | | medium | [5, 12) (365) | [12, 15] (95) |
| | | hard | [5, 11) (342) | [11, 15] (118) |

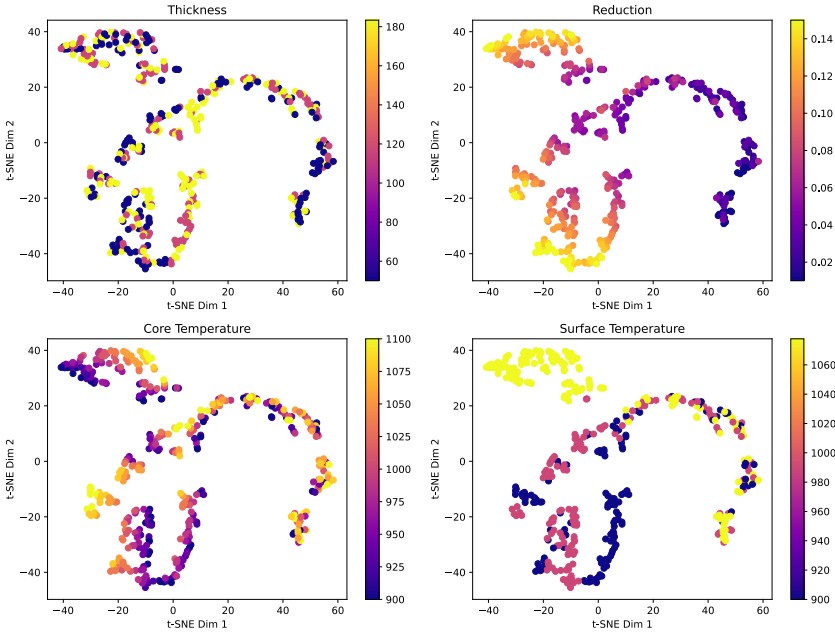

Figure 7: T-SNE visualization of the conditioning vectors for the *hot rolling* dataset. Point color indicates the magnitude of the respective parameter. While the sheet thickness $t$ appears to be uniformly distributed, the remaining three exhibit distinct clustering patterns. Taking into account domain knowledge from industry experts, we defined the reduction parameter $r$ as the basis for constructing distribution shifts.

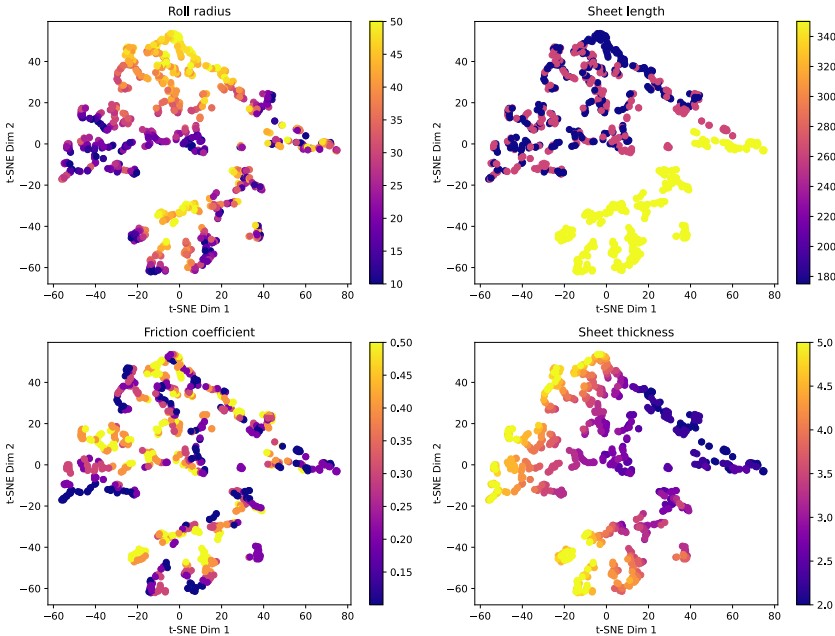

Figure 8: T-SNE visualization of the conditioning vectors for the *sheet metal forming* dataset. Point color indicates the magnitude of the respective parameter. The sheet length $l$ shows the most distinct groupings, but with only three discrete values, it is unsuitable for defining domain splits. The friction coefficient $\mu$ appears uniformly distributed across the embedding. In contrast, sheet thickness $t$ and roll radius $r$ show clustering behavior, making them more appropriate candidates for inducing distribution shifts. We choose $t$ as the domain defining parameter.

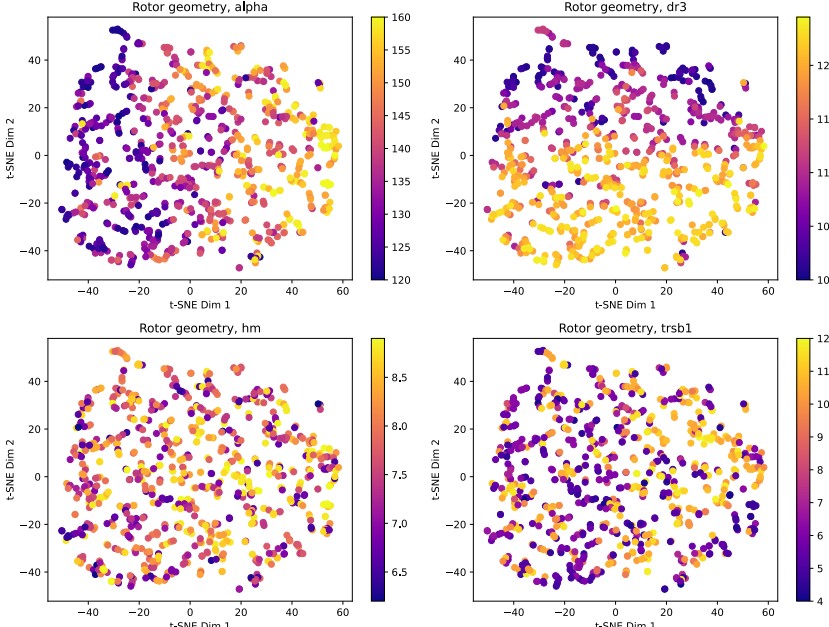

Figure 9: T-SNE visualization of the conditioning vectors for the *electric motor design* dataset. Point color indicates the magnitude of the respective parameter. For clarity, we only show selected parameters. The only parameter for which exhibits see some structure in the latent space is $d_{r3}$, we therefore choose this to be our domain defining parameter in accordance with domain experts.

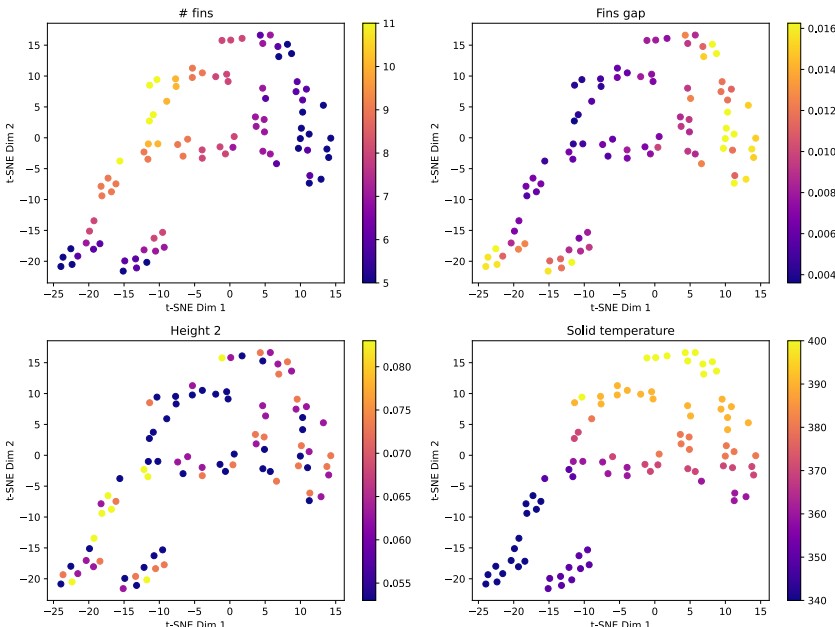

Figure 10: T-SNE visualization of the conditioning vectors for the *heatsink design* dataset. Point color indicates the magnitude of the respective parameter. Height 2 is distributed equally across the representation, but the other parameters show concrete grouping behavior. We therefore choose the number of fins as the domain defining parameter.

## C  Model Architectures

This section provides explanations of all model architectures used in our benchmark. All models are implemented in PyTorch and are adapted to our conditional regression task. All models have in common, that they take node coordinates as inputs and embed them using a sinusoidal positional encoding. Additionally, all models are conditioned on the input parameters of the respective simulation sample, which are encoded through a conditioning network described below.

**Conditioning Network.**  The conditioning module used for all neural surrogate architectures embeds the simulation input parameters into a latent vector used for conditioning. The network consists of a sinusoidal encoding followed by a simple MLP. The dimension of the latent encoding is 8 throughout all experiments.

**PointNet.**  Our PointNet implementation is adapted from [96] for node-level regression. Input node coordinates are first encoded using sinusoidal embeddings and passed through an encoder MLP. The resulting representations are aggregated globally using max pooling over nodes to obtain a global feature vector. To propagate this global feature, it is concatenated back to each point's feature vector. This fused representation is then fed into a final MLP, which produces the output fields. The conditioning is performed by concatenating the conditioning vector to the global feature before propagating it to the nodes features. We use a PointNet base dimension of 16 for the small model and 32 for the larger model.

**GraphSAGE.**  We adapt GraphSAGE [79] to the conditional mesh regression setting. Again, input node coordinates are embedded using a sinusoidal encoding and passed through an MLP encoder. The main body of the model consists of multiple GraphSAGE message passing layers with mean aggregation. We support two conditioning modes, namely concatenating the latent conditioning vector to the node features, or applying FiLM style modulation [102] to the node features before each message passing layer. We always use FiLM modulation in the presented results. After message passing, the node representations are passed through a final MLP decoder to produce the output fields. The base dimension of the model is kept at 128 and we employ 4 GraphSAGE layers.

**Transsolver.** The Transsolver model follows the originally introduced architecture [101]. Similar to the other models, node coordinates first are embedded using a sinusoidal encoding and passed through an MLP encoder to produce initial features. Through learned assignement, each node then gets mapped to a slice, and inter- as well as intra-slice attention is performed. Afterwards, fields are decoded using an MLP readout. The architecture supports two conditioning modes: concatenation, where the conditioning vector is concatenated to the input node features before projection, or modulation through DiT layers across the network. For our experiments, DiT is used. We choose a latent dimension of 128, a slice base of 32 and we apply four attention blocks for the small model. For the larger model, we scale to 256, 128 and 8 layers respectively.

**UPT.** Our UPT implementation builds on the architecture proposed in [73]. First, a fixed number of supernodes are uniformly sampled from the input nodes. Node coordinates are embedded using a sinusoidal encoding followed by an MLP. The supernodes aggregate features from nearby nodes using one-directional message passing and serve as tokens for subsequent transformer processing. They are then processed by stack of DiT blocks, which condition the network on the simulation input parameters. For prediction, we employ a DiT Perceiver [104] decoder that performs cross-attention between the latent representation and a set of query positions. This allows the model to generate field predictions at arbitrary spatial locations, which is a desirable property for inference. We sample 4096 supernodes and use a base dimension of 192. We use 8 DiT blocks for processing and 4 DiT Perceiver blocks for decoding.

# D Experiments

This section provides a detailed overview of the performed experiments for this benchmark. First, we explain the benchmarking setup used to generate the benchmarking results in detail in Appendix D.1 and the evaluation procedure in Appendix D.2. Furthermore, we provide information about training times for the presented methods in Appendix D.3.

## D.1 Experimental Setup

**Dataset Splits.** We split each dataset into source and target domains as outlined in Section 3.5 and Appendix B. Within source domains, we use a 50%/25%/25% split for training, validation, and testing, respectively. For target domains, where labels are unavailable during training in our UDA setup, we use a 50%/50% split for training and test sets. The large validation and test sets are motivated the industrial relevance of our benchmark, where reliable performance estimation on unseen data is a crucial factor.

**Training Pipeline.** For training, we use a dataset wide per field z-score normalization strategy, with statistics computed on the source domain training set. We use a batch size of 16 and the AdamW optimizer [105] with a weight decay of 1e-5 and a cosine learning rate schedule, starting from 1e-3. Gradients are clipped to a maximum norm of 1. For the large scale *heatsink design* dataset, we enable Automatic Mixed Precision (AMP) to reduce memory consumption and training time. Additionally, we use Exponential Moving Average (EMA) updates with a decay factor of 0.95 to stabilize training.

Performance metrics are evaluated every 10 epochs, and we train all models for a maximum of 3000 epochs with early stopping after 500 epochs of no improvement on the source domain validation loss.

**Domain Adaptation Specifics.** To enable UDA algorithms, we jointly sample mini batches from the source and target domains at each training step and pass them thorugh the model. Since target labels are not available, we compute supervised losses only on the source domain outputs. In addition, we compute DA losses on the latent representations of source and target domains in order to encourage domain invariance.

Since a crucial factor in the performance of UDA algorithms is the choice of the domain adaptation loss weight $\lambda$, we perform extensive sweeps over this hyperparameter and select models using the unsupervised model selection strategies described in Section 4.3.

For the three smaller datasets, we sweep $\lambda$ logarithmically over $\lambda \in \{10^{-1}, 10^{-2}, \ldots, 10^{-9}\}$, while for the large scale *Heatsink design* dataset, we sweep a smaller range, namely $\lambda \in \{10^2, 10^{-1}, \ldots, 10^{-2}\}$, motivated by the balancing principle [57].

Table 8 provides an overview of the number of trained models for benchmarking performance of all models and all UDA algorithms on the *medium* difficulty domain shifts across all datasets.

Table 8: Overview of the benchmarking setup and number of trained models across all datasets.

| Dataset | Models | UDA algorithms | $\lambda$ values | # seeds | # models trained |
|---------|--------|----------------|------------------|---------|------------------|
| Rolling | PointNet, GraphSAGE, Transolver | Deep Coral, CMD, DANN | $\{10^{-1}; 10^{-9}\}$ | 4 | 324 |
| | | w/o UDA | $-$ | 4 | 12 |
| Forming | PointNet, GraphSAGE, Transolver | Deep Coral, CMD, DANN | $\{10^{-1}; 10^{-9}\}$ | 4 | 324 |
| | | w/o UDA | $-$ | 4 | 12 |
| Motor | PointNet, GraphSAGE, Transolver | Deep Coral, CMD, DANN | $\{10^{-1}; 10^{-9}\}$ | 4 | 324 |
| | | w/o UDA | $-$ | 4 | 12 |
| Heatsink | PointNet, Transover, UPT | Deep Coral, CMD, DANN | $\{10^{2}; 10^{-2}\}$ | 4 | 180 |
| | | w/o UDA | $-$ | 4 | 12 |
| **Sum** | | | | | **1,200** |

**Additional Details.** For the three smaller datasets, we use smaller networks, while for the large scale *heatsink design* dataset, we train larger model configurations to accommodate the increased data complexity. An overview of model sizes along with average training times per dataset is provided in Table 9. We also refer to the accompanying code repository for a complete listing of all model hyperparameters, where we provide all baseline configuration files and detailed step by step instructions for reproducibility of our results.

Another important detail is that, during training on the *heatsink design* dataset, we randomly subsample 16,000 nodes from the mesh in each training step to ensure computational tractability. However, all reported performance metrics are computed on the full resolution of the data without any subsampling.

## D.2 Evaluation Metrics

We report the RMSE for each predicted output field. For field $i$, the RMSE is defined as:

$$\text{RMSE}_i^{\text{field}} = \frac{1}{M} \sum_{m=1}^{M} \sqrt{\frac{1}{N_m} \sum_{n=1}^{N_m} \left( y_{m,n}^{(i)} - f(x)_{m,n}^{(i)} \right)^2},$$

where $M$ is the number of test samples (graphs), $N_m$ the number of nodes in graph $m$, $y_{m,n}^{(i)}$ the ground truth value of field $i$ at node $n$ of graph $m$, and $f(x)_{m,n}^{(i)}$ the respective model prediction.

For aggregated evaluation, we define the total Normalized RMSE (NRMSE) as:

$$\text{NRMSE} = \sum_{i=1}^{K} \text{NRMSE}_i^{\text{field}},$$

where $K$ is the number of predicted fields. For this metric, all individual field errors are computed on normalized fields before aggregation.

In addition to the error on the fields, we report the mean Euclidean error of the predicted node displacement. This is computed based on the predicted coordinates $\hat{\mathbf{c}}_{m,n} \in \mathbb{R}^d$ and the ground truth coordinates $\mathbf{c}_{m,n} \in \mathbb{R}^d$, where $d \in \{2, 3\}$ is the spatial dimensionality, as follows:

$$\text{RMSE}^{\text{deformation}} = \frac{1}{M} \sum_{m=1}^{M} \frac{1}{N_m} \sum_{n=1}^{N_m} \|\mathbf{c}_{m,n} - \hat{\mathbf{c}}_{m,n}\|_2.$$

## D.3 Computational Resources and Timings

While generating the results reported on the *medium* difficulty level of our benchmark, we measured average training times per dataset and model architecture. While the total compute budget is difficult to estimate due to distributed training runs across various hardware setups, we report standardized average training times for 2000 epochs in Table 9, measured on a single NVIDIA H100 GPU using batch size of 16.

Table 9: Average training times (averaged for 2000 epochs) and parameter counts for each model on the *medium* difficulty benchmark tasks. Times are measured on a H100 GPU using a batch size of 16.

| Dataset | Model | # parameters | Avg. training time (h) |
|---------|-------|--------------|------------------------|
| Rolling | PointNet | 0.3M | 1.2 |
| | GraphSAGE | 0.2M | 3 |
| | Transolver | 0.57M | 2.1 |
| Forming | PointNet | 0.3M | 2.8 |
| | GraphSAGE | 0.2M | 8 |
| | Transolver | 0.57M | 4.4 |
| Motor | PointNet | 0.3M | 5.6 |
| | GraphSAGE | 0.2M | 11.5 |
| | Transolver | 0.57M | 6.5 |
| Heatsink | PointNet | 1.08M | 4.9 |
| | Transolver | 4.07M | 5.3 |
| | UPT | 5.77M | 5.5 |

## E   Dataset Details

### E.1   Hot Rolling

Table 10: Input parameter ranges for the *hot rolling* simulations. Samples are generated by equally spacing each parameter within the specified range using the indicated number of steps, resulting in $5 \times 19 \times 10 \times 5 = 4750$ total samples.

| Parameter | Description | Min | Max | Steps |
|-----------|-------------|-----|-----|-------|
| $t\ (mm)$ | Initial slab thickness. | 50.0 | 183.3 | 5 |
| reduction $(-)$ | Reduction of initial slab thickness. | 1.0 | 15.0 | 19 |
| $T_{\text{core}}\ (^{\circ}C)$ | Core slab temperature. | 900.0 | 1000.0 | 10 |
| $T_{\text{surf}}\ (^{\circ}C)$ | Surface slab temperature. | 900.0 | 1077.77 | 5 |

### E.2   Sheet Metal Forming

Table 11: Input parameter ranges for the *sheet metal forming* simulations. Samples are generated by equally spacing each parameter within the specified range using the indicated number of steps, resulting in $17 \times 13 \times 3 \times 5 = 3315$ total samples.

| Parameter | Description | Min | Max | Steps |
|-----------|-------------|-----|-----|-------|
| $r\ (mm)$ | Roll radius. | 10.0 | 50.0 | 17 |
| $t\ (mm)$ | Sheet thickness. | 2.0 | 5.0 | 13 |
| $l\ (mm)$ | Sheet length. | 175.0 | 350.0 | 3 |
| $\mu\ (-)$ | Friction coefficient between... | 0.1 | 0.5 | 5 |

 **E.3   Electric Motor Design**

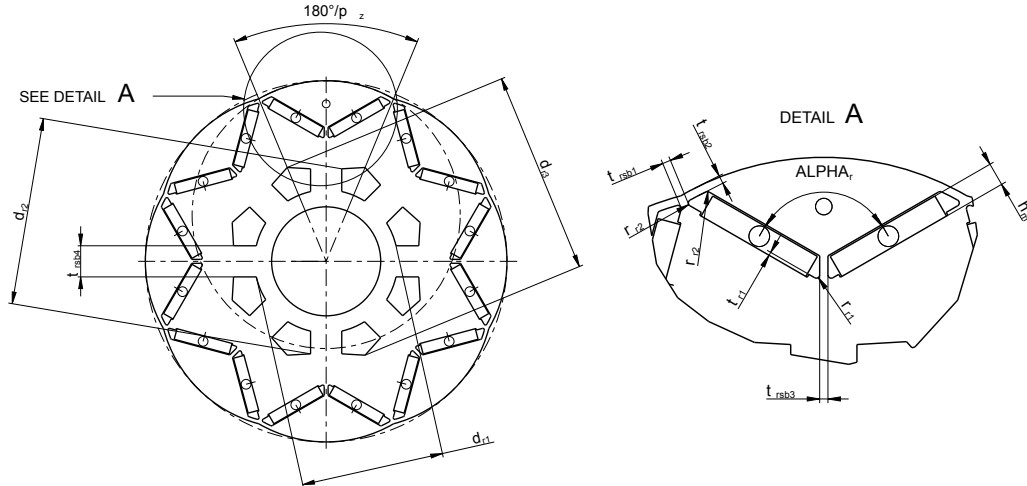

Figure 11: Technical drawing of the electrical motor. Sampling ranges for the shown parameters can be found in Table 12.

Table 12: Input parameters for the *electric motor design* simulations. Since this simulation was performed by domain experts, the parameters are not uniformly sampled as in the previous simulation scenarios. In total, 3196 simulations were performed.

| Parameter | Description | Min | Max |
|---|---|---|---|
| $d_{si}$ $(mm)$ | Stator inner diameter. | 150.0 | 180.0 |
| $h_m$ $(mm)$ | Magnet height. | 6.0 | 9.0 |
| $\alpha_r$ $(°)$ | Angle between magnets. | 120.0 | 160.0 |
| $t_{r1}$ $(mm)$ | Magnet step. | 1.0 | 5.0 |
| $r_{r1}$ $(mm)$ | Rotor slot fillet radius 1. | 0.5 | 2.5 |
| $r_{r2}$ $(mm)$ | Rotor slot fillet radius 2. | 0.5 | 3.5 |
| $r_{r3}$ $(mm)$ | Rotor slot fillet radius 3. | 0.5 | 5.0 |
| $r_{r4}$ $(mm)$ | Rotor slot fillet radius 4. | 0.5 | 3.0 |
| $t_{rsb1}$ $(mm)$ | Thickness saturation bar 1. | 4.0 | 12.0 |
| $t_{rsb2}$ $(mm)$ | Thickness saturation bar 2. | 1.0 | 3.0 |
| $t_{rsb3}$ $(mm)$ | Thickness saturation bar 3. | 1.2 | 4.0 |
| $t_{rsb4}$ $(mm)$ | Thickness saturation bar 4. | 5.0 | 12.0 |
| $d_{r1}$ $(mm)$ | Rotor slot diameter 1. | 60.0 | 80.0 |
| $d_{r2}$ $(mm)$ | Rotor slot diameter 2. | 80.0 | 120.0 |
| $d_{r3}$ $(mm)$ | Rotor slot diameter 3. | 100.0 | 125.0 |

 **E.4  Heatsink Design**

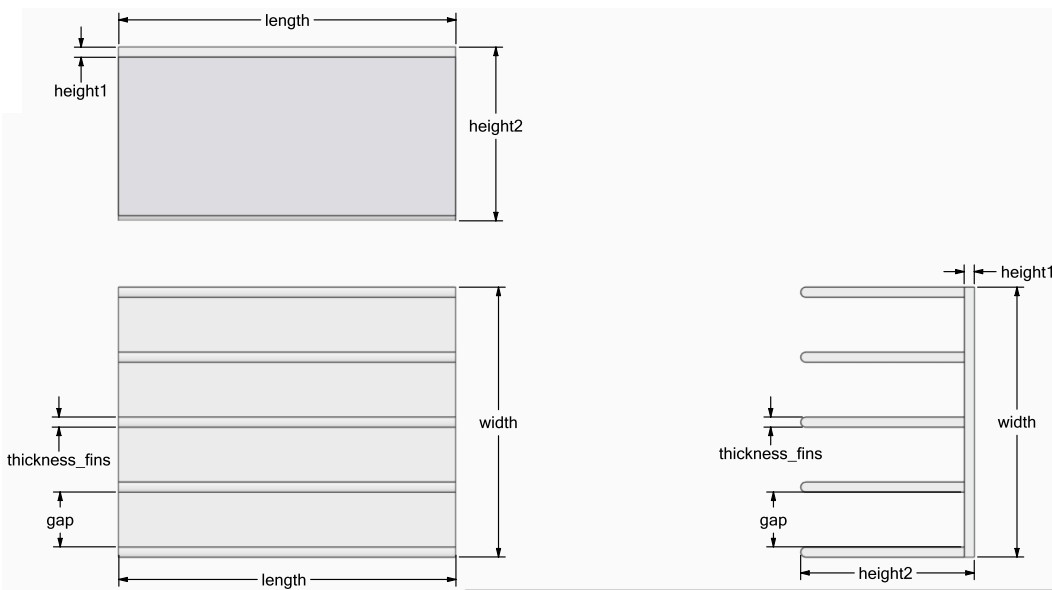

Figure 12: Technical drawing of the solid body in the *heatsink design* dataset. Some of the shown parameters are varied for data generation (see Table 13).

Table 13: Input parameters for the *heatsink design* simulations. The simulation was performed by domain experts and the parameters are not uniformly sampled as in the previous simulation scenarios. In total, 460 simulations were performed.

| Parameter | Description | Min | Max |
|---|---|---|---|
| fins $(-)$ | Number of fins. | 5 | 14 |
| gap $(m)$ | Gap between fins. | 0.0023 | 0.01625 |
| height2 $(m)$ | Height 2. | 0.053 | 0.083 |
| T (solid) $(K)$ | Temperature of the solid fins. | 340 | 400 |

