# OpenReview forum: "SIMSHIFT: A Benchmark for Adapting Neural Surrogates to Distribution Shifts"
_NeurIPS.cc/2025/Datasets_and_Benchmarks_Track — Submitted to NeurIPS 2025 Datasets and Benchmarks Track_

### Official Review · Reviewer_ErFp · 2025-06-25

**Rating:** 4
**Confidence:** 1

**Summary:**

Neural surrogates for PDEs often struggle to generalize to unseen configurations, such as new materials or geometries. To address this issue, the authors introduce SIMSHIFT, a benchmark suite comprising four industrial simulation tasks. They also adapt domain adaptation (DA) techniques to neural surrogates, enabling prediction on target configurations using only parametric descriptions without access to ground truth simulations. Experiments on SIMSHIFT demonstrate the challenges of out-of-distribution generalization, highlight the potential of DA in simulation, and reveal open questions in building robust surrogates under distribution shifts.

**Dataset Code Accessibility:**

Yes

**Ethical Considerations:**

No, there are no or only very minor ethics concerns

**Limitations Weaknesses:**

1.	The datasets are restricted to steady-state problems, excluding time-dependent PDEs, which are increasingly important in many real-world applications.
2.	The domain shifts are defined only through scalar parameter variations, neglecting more complex and realistic shifts such as changes in mesh geometry or topology.
3.	The benchmarks focus on variations in a single parameter at a time, which may not fully reflect the complexity of real-world simulation scenarios that involve multi-factor domain shifts.

**Strengths Contributions:**

1.	The paper introduces SIMSHIFT, a valuable and practical benchmark dataset collection for evaluating unsupervised domain adaptation (UDA) methods and neural operators in real-world physical simulation scenarios.
2.	The authors conduct extensive experiments on physical simulation data, offering an in-depth assessment of UDA techniques and their effects on neural surrogate performance.
3.	Results show that standard UDA training methods can significantly improve generalization of neural operators to unseen parameter ranges—consistent with improvements reported in the broader UDA literature.
4.	The study highlights the critical role of unsupervised model selection, showing it can have as much impact on downstream performance as UDA training itself.

---

> ### Author Rebuttal · Authors · 2025-07-31
>
> We thank reviewer ErFp for the invested time and their feedback, raising valuable points. Below, we address each listed limitation.
>
> ### 1. The datasets are restricted to steady-state problems, excluding time-dependent PDEs.
> We appreciate this observation, which was also raised reviewer 2uJ6.
>
> We intentionally design our benchmark around steady-state problems for two main reasons:
>
> __1. Industrial relevance:__ In general, many engineering workflows and manufacturing applications explicitly focus on steady-state or time-averaged solutions rather than instantaneous dynamics since time scales of interest are much longer than the fluctuation time scales in transient solutions. In most of CFD contexts, it is not the instantaneous velocity field but the mean flow and its associated transport properties (e.g., drag, heat transfer, mixing rates) that determine key performance metrics. That's why methods like Reynolds-averaged Navier–Stokes (RANS) and time-averaged Large Eddy Simulations (LES) are widely adopted in industries such as aerospace and automotive design optimization. Other well-enstablished engineering datasets such as DrivAerNet++ [1] and DrivAerML [2] also use the steady-state formulation. Both focus on car design optimization and are among the largest, most expensive and industry-relevant publicly available datasets in machine learning today.
>
> __2. Benchmarking clarity:__ Steady-state scenarios avoid the additional complexities arising from autoregressive error accumulation. This makes benchmarking cleaner, because some accumulating error effects could be amplified by domain shifts. Steady rollout and the evaluation of neural surrogates for such systems is still an area of research in the community, since there are no standardized evaluation metrics and suitable measures differ for different systems.
>
> We acknowledge that transient simulations can also be important for industrial applications, and future extensions of SIMSHIFT will consider incorporating these scenarios. However, we believe the current steady-state focus provides a valuable and justified starting point for benchmarking UDA methods for neural surrogates in settings with industrial relevance.
>
> [1] Ashton, N, et al. "DrivAerML: High-Fidelity Computational Fluid Dynamics Dataset for Road-Car External Aerodynamics." preprint, 2024.
>
> [2] Elrefaie M. et al. "DrivAerNet++: A Large-Scale Multimodal Car Dataset with Computational Fluid Dynamics Simulations and Deep Learning Benchmarks." NeurIPS, 2024.
>
> ### 2. The domain shifts are defined only through scalar parameter variations, neglecting more complex and realistic shifts such as changes in mesh geometry or topology.
> Thank you for bringing up this consideration, which was also raised by reviewer oE7g.
>
> Effectively, in 3 of the 4 presented datasets, __the domain shift defining parameter(s) describe the geometry of the samples__ (see Table 7 together with Figures 3a, 11 and 12); meaning that the geometry is parametrized by a set of scalars.
>
> The idea behind our conditioning setup is to allow plug-in integration of any model architecture and domain adaptation method.
>
> In initial ablations of our benchmark we experimented with mesh encoder conditioning networks instead of explicitly conditioning on the scalar parameters. It is worth noting that this choice is not architecture agnostic.
>
> Below, we include results on the motor dataset of a PointNet architecture with a geometric conditioner and without explicit access to the scalar parameters. We follow the procedure described in our paper: we train the model with each UDA algorithm and 9 different regularizer strengths, over 4 seeds (mean shown in results below). Values in the parentheses show improvement to the unregularized models.
>
> | UDA method | Best model selection | All fields normalized averaged (-) | Mises Stress (MPa) |
> | --- | --- | ---- | ---- |
> | DANN | SB | 0.40 (−0.07) | 32.96 (−3.81) |
> | CMD | SB | 0.39 (−0.07) | 32.79 (−3.99) |
> | Deep Coral | SB | 0.38 (−0.09) | 31.25 (−5.53) |
> | Oracle (Deep Coral) | TB | 0.38 (−0.09) | 31.25 (−5.53) |
>
> While UDA consistently improves target error, the errors are higher compared to our proposed benchmark setup (see Mises Stress in Table 2 and all fields normalized avg Table 5). Because of worse performance and less trivial extension to other architectures, we decided to go for the conditioning MLP in our final design (see Figure 1).
>
> ### 3. The benchmarks focus on variations in a single parameter at a time.
> This is an excellent point, which also raised by reviewer oE7g.
>
> We agree that multi-dimensional shifts present a more challenging scenario. To address this, we add preliminary results on __2-dimensional shifts__ in the electric motor design dataset. In more detail, we jointly shift the rotor slot diameter $d_{r3}$ (parameter shifted in the cases from the paper) and the angle between the magnets $\alpha$. We split the domains as follows:
>
> | Parameter | Source range | Target range |
> | --- | --- | --- |
> | Rotor slot diameter $d_{r3}$ | [99, 120) | [120, 130] |
> | Angle between magnets $\alpha$ | [119, 153) | [153, 170]
>
> We trained 336 additional models on this configuration, and present a summary of the results in the table below:
>
> | All Models | Best UDA method | Best model selection | All fields normalized avg (-) | Mises Stress (MPa) |
> | --- | --- | --- | --- | --- |
> | PointNet | DANN | SB | 0.35 (−0.09) | 28.84 (−5.02) |
> | GraphSAGE | Deep Coral | IWV | 0.33 (−0.05) | 28.27 (−1.39) |
> | Transolver | DANN | SB | 0.11 (−0.04) |  8.36 (−2.01) |
> | Oracle (Transolver) | CMD | TB | 0.11 (−0.04) |  8.30 (−2.06) |
>
> The table shows the best performing UDA algorithm and unsupervised model selection combination per architecture. Again, we report mean error values on the target domain accross 4 seeds and the improvement towards the corresponding unregularized baseline models in parentheses. Comparing these results with the original 1-dimensional shift (Table 2), we highlight two aspects:
>
> - __Increased error__: for 3 out of 4 architectures, the average error in Mises Stress is consistently larger than in the 1D shift setting, confirming that the 2D shift is indeed more challenging.
> - __Greater benefits from UDA__: the (relative) improvements compared to the unregularized baseline suggest that domain adaptation training is more beneficial, for this particular case.

---

### Official Review · Reviewer_2uJ6 · 2025-07-01

**Rating:** 4
**Confidence:** 3

**Summary:**

The paper introduces SIMSHIFT, a benchmark for evaluating domain adaptation (DA) methods in neural PDE surrogates under distribution shifts, featuring four industrial simulation tasks (hot rolling, sheet metal forming, electric motor design, heatsink design) with predefined shifts of varying difficulty. It extends DA techniques (e.g., Deep-Coral, CMD, DANN) to neural operators (PointNet, GraphSAGE, Transolver) and evaluates unsupervised model selection strategies (IWV, DEV), demonstrating performance gains through 1,200 experiments while highlighting gaps in current approaches. The modular benchmark suite enables standardized testing, with limitations including steady-state focus and single-parameter shifts, suggesting future work on dynamic systems and complex distribution shifts.

**Additional Feedback:**

I am willing to improve the rating if the issues are well solved.

**Dataset Code Accessibility:**

Yes

**Dataset Code Comments:**

The dataset is provided in a Hugging Face link.

**Ethical Considerations:**

No, there are no or only very minor ethics concerns

**Final Justification:**

The authors provide a clear rationale for current design choices, address key concerns partly, and present a well-motivated benchmark with strong potential for impactful future extensions.

**Limitations Weaknesses:**

1. The benchmark only covers time-independent PDEs (Sec. 3, Sec. 6), omitting critical industrial applications like transient fluid dynamics or autoregressive error accumulation in time-dependent systems (Sec. 3, Footnote 84).
2. The easy/medium/hard splits are heuristic; no theoretical justification (e.g., Wasserstein distance between domains) is provided.
3. Overloaded Tables: Tables 3–6 (Appendix A) are dense and hard to parse; key trends (e.g., DA vs. no-DA) are buried.

**Strengths Contributions:**

1. This paper proposes the first benchmark to systematically study DA for neural surrogates on unstructured meshes, and introduces multi-difficulty shifts (easy/medium/hard) and unsupervised model selection strategies.
2. The evaluation is extensive, with 1,200 runs across 4 datasets, 3 DA methods, and 4 selection strategies.
3. The presentation is well-structured, with a clear problem formulation and experimental setup.

---

> ### Author Rebuttal · Authors · 2025-07-31
>
> We appreciate reviewer 2uJ6’s comments, which raise several important points. Please find our answers below.
>
> ### 1. The benchmark only covers time-independent PDEs.
> Thank you for bringing this up, as also noted by reviewer ErFp.
>
> We intentionally design our benchmark around steady-state problems for two main reasons:
>
> __1. Industrial relevance:__ In general, many engineering workflows and manufacturing applications explicitly focus on steady-state or time-averaged solutions rather than instantaneous dynamics since time scales of interest are much longer than the fluctuation time scales in transient solutions. In most of CFD contexts, it is not the instantaneous velocity field but the mean flow and its associated transport properties (e.g., drag, heat transfer, mixing rates) that determine key performance metrics. That's why methods like Reynolds-averaged Navier–Stokes (RANS) and time-averaged Large Eddy Simulations (LES) are widely adopted in industries such as aerospace and automotive design optimization. Other well-enstablished engineering datasets such as DrivAerNet++ [1] and DrivAerML [2] also use the steady-state formulation. Both focus on car design optimization and are among the largest, most expensive and industry-relevant publicly available datasets in machine learning today.
>
> __2. Benchmarking clarity:__ Steady-state scenarios avoid the additional complexities arising from autoregressive error accumulation. This makes benchmarking cleaner, because some accumulating error effects could be amplified by domain shifts. Steady rollout and the evaluation of neural surrogates for such systems is still an area of research in the community, since there are no standardized evaluation metrics and suitable measures differ for different systems.
>
> We acknowledge that transient simulations can also be important for industrial applications, and future extensions of SIMSHIFT will consider incorporating these scenarios. However, we believe the current steady-state focus provides a valuable and justified starting point for benchmarking UDA methods for neural surrogates in settings with industrial relevance.
>
> [1] Ashton, N, et al. "DrivAerML: High-Fidelity Computational Fluid Dynamics Dataset for Road-Car External Aerodynamics." preprint, 2024.
>
> [2] Elrefaie M. et al. "DrivAerNet++: A Large-Scale Multimodal Car Dataset with Computational Fluid Dynamics Simulations and Deep Learning Benchmarks." NeurIPS, 2024.
>
> ### 2. The easy/medium/hard splits are heuristic; no theoretical justification.
> We sincerely thank reviewer 2uJ6 for bringing this up. This is a very valid concern. Besides empirical inputs of domain experts from practice, the source and target splits for the different difficulties have been set based on initial experimentation, where we compared differences in model errors between the domains -- which is often called *transfer error*. The maximum transfer error (sup over some function class) can be bounded (see, e.g., [3, 4] for a summary) by distances such as the H-divergence (estimate is poxy A-distance (PAD) [5, 6]), the Maximum Mean Discrepancy and Wasserstein distances.
>
> To strengthen our analysis, we additionally provide the PAD in the output space (ground truth simulation fields), where we rely on a PointNet mesh source/target classifier for PAD estimation. We show the results for all three difficulty levels and all four datasets in the following table:
>
> | Dataset     |  Easy | Medium |  Hard |
> | ---------- | ---- | ----- | ---- |
> | **Rolling** | 1.063 |  1.159 | 1.210 |
> | **Forming** | 0.860 |  0.938 | 1.030 |
> | **Motor**   | 0.762 |  0.932 | 0.955 |
> | **Heatsink**   | 1.446 |  1.683 | 1.861 |
>
> The results are consistent with our paper (transfer error estimation, Figure 6): For all datasets, __PAD increases with increasing difficulty.__
>
> [3] Bouvier et al. "Robust domain adaptation: Representations, weights and inductive bias." ECML, 2020.
>
> [4] Johansson et al. "Support and invertibility in domain-invariant representations." AISTATS 2019.
>
> [5] Zellinger et al. "The balancing principle for parameter choice in distance-regularized domain adaptation." NeurIPS, 2021.
>
> [6] Ben-David et al. "A theory of learning from different domains." Machine learning, 2010.
>
> ### 3. Overloaded Tables: Tables 3–6 (Appendix A) are dense and hard to parse; key trends (e.g., DA vs. no-DA) are buried.
> Admittedly, the overview tables in the Appendix contain a large amount of information. This is because they aim to display, for each dataset, all combinations of models, UDA algorithms and model selection strategies. Their purpose is reproducibility and transparency, although it is true that they can be hard to navigate.
>
> Recognizing the complexity of navigating such detailed results, we included Table 2 in the main body of the paper, where we summarize _DA vs. no-DA_ trends to convey them clearly (DA promotion indicated in parentheses after each result).
>
> On that note, we will try to increase the clarity of the tables in the Appendix by color-coding and highlighting the best performing combination for every model, UDA and model selection in the tables in the Appendix in the next revision of the paper.

---

> > ### Comment · Reviewer_2uJ6 · 2025-08-05
> >
> > Thank you for the additional clarifications in response to both my review and the AC’s high-level questions. I appreciate the more concrete industrial examples, the explicit link to regulatory requirements, and the further details on the rationale for the chosen domain shift design. The following points summarize my evaluation:
> >
> > **Importance of distribution shift in industrial contexts**
> > The real-world scenarios provided (e.g., novel geometries, material changes, sensor drift) and the connection to compliance frameworks (EU AI Act) help to ground the benchmark’s motivation. This addresses my earlier request for a more concrete and compelling justification.
> >
> > **Shift design (univariate vs. multivariate)**
> > I acknowledge the rationale for starting with explicit, univariate parameter shifts for controllability and interpretability, and the addition of 2D shift experiments is a useful extension. However, the benchmark would be more representative if some multivariate or implicit data-space shifts were included earlier, even in limited form, as these may better reflect realistic industrial variations.
> >
> > **Time-independent PDEs**
> > The justification for focusing on steady-state or time-averaged solutions is clear and well supported by references, especially for design-stage optimization workflows. That said, transient scenarios remain relevant in many domains (e.g., weather, certain mechanical systems), and even a small-scale pilot inclusion would broaden the scope and adoption potential.
> >
> > **Remaining opportunities**
> > The work would benefit from: (a) gradually integrating more complex shift types beyond univariate/2D parameter gaps, and (b) introducing at least partial transient-PDE evaluation. These additions could significantly enhance the generality of the benchmark without undermining its current focus and clarity.
> >
> > Overall, the additional response strengthens the motivation and design rationale, and the benchmark’s industrial relevance is now clearer. The current scope is reasonable for a first release, but expanding shift diversity and PDE coverage in future versions would further increase its scientific and practical impact.

---

> > > ### Author Response · Authors · 2025-08-05
> > >
> > > Thank you for engaging in the discussion. We are already running the training on the multi-dimensional shifts (we started with 2D motor so we could include preliminary results in the rebuttal). We will include the full results for the more challenging multi-dimensional shifts in the final version.
> > >
> > > What exactly do you mean with implicit data-space shifts? Do you mean the geometry implicit setting (like in rebuttal for reviewer oE7g, question 3a)?

---

> > > > ### Comment · Reviewer_2uJ6 · 2025-08-05
> > > >
> > > > By implicit data-space shifts, I mean distribution changes not explicitly parameterized by a small set of known scalars in the input space, but arising from alterations in the data distribution itself—for example, free-form geometry changes not reducible to a few shape parameters, mesh resolution or discretization differences, sensor drift, or correlated multi-parameter variations. The “geometry implicit” setting mentioned in the rebuttal for reviewer oE7g (3a) is one instance of such a shift, but my usage is broader and includes any change that modifies the data distribution without relying on a predefined low-dimensional parametric description.
> > > >
> > > > Such implicit shifts are important for benchmark representativeness because real-world industrial distribution changes are rarely driven by a single explicit parameter. They often emerge from multi-factor, correlated, and sometimes unobservable variations—for example, manufacturing tolerances interacting with material properties, or solver/meshing changes in simulation pipelines. Benchmarks limited to simple, explicit parametric gaps may overestimate the robustness of UDA methods, as these methods could exploit parameter-specific cues that do not generalize to more complex, implicit shifts. Including such scenarios would improve ecological validity and better stress-test algorithms under realistic and challenging conditions.

---

> > ### Author Response · Authors · 2025-08-06
> >
> > Dear Reviewer 2uJ6,
> >
> > Thank you for the clarification. While exploring shifts based directly on changes in the data distribution is an interesting direction and we provided some results in this direction in our rebuttal (point 2), it is non-trivial in our setting and deserves a dedicated publication of its own. We therefore disagree that this benchmark would benefit from including shifts defined on the data-space since it is too complex (technically and computationally) to decently control these shifts, making it impossible to provide justified shift scenarios.
> >
> > Although we acknowledge the possibility to make the benchmark even more real-world like, we believe taking this step too early would even stop researchers from studying these problems in neural simulation since there is no prior work in this direction. __Don’t you think introducing less structured or more loosely defined shifts would hinder adoption and development in the field?__ To emphasise again, this is to the best of our knowledge the first benchmark for multi-dimensional regression problems in DA, given that the classical datasets such as Office Home [1], or DomainNet [2] dominated the field for years. With this work we try to encourage a shift in the DA community to move away from such outdated classification datasets, and at the same time encourage engagement with DA in the neural simulation community, by providing well defined and accessible problems.
> >
> > Best,
> >
> > Authors
> >
> > [1] Venkateswara, H., Eusebio, J., Chakraborty, S., & Panchanathan, S. (2017). Deep hashing network for unsupervised domain adaptation. In Proceedings of the IEEE Conference on Computer Vision and Pattern Recognition (pp. 5018–5027).
> >
> > [2] Peng, X., Bai, Q., Xia, X., Huang, Z., Saenko, K., & Wang, B. (2019). Moment matching for multi-source domain adaptation. In Proceedings of the IEEE International Conference on Computer Vision (pp. 1406–1415).

---

> > > ### Comment · Reviewer_2uJ6 · 2025-08-08
> > >
> > > Thank you for clarifying the challenges of incorporating implicit data-space shifts. I agree that prioritizing controlled, explicit shifts makes sense for this release. Still, briefly outlining possible future extensions toward more complex shifts could inspire follow-up work and strengthen the benchmark’s long-term impact. Overall, the work is well-motivated, and I am inclined to raise my score to 4.

---

> > > > ### Author Response · Authors · 2025-08-08
> > > >
> > > > Thank you for your considerations. We will make sure to include results on the data-space distance of the splits as well as outlining the discussed future directions clearly in the final version.
> > > >
> > > > Best,
> > > >
> > > > Authors

---

### Official Review · Reviewer_oE7g · 2025-07-01

**Rating:** 4
**Confidence:** 3

**Summary:**

This paper presents SIMSHIFT, a benchmark suite for Unsupervised Domain Adaptation (UDA) applied to neural PDE surrogates across four industrial simulation tasks: hot rolling, sheet metal forming, electric motor design, and heatsink design. The benchmark includes parametric domain shifts, baseline neural operators (PointNet, GraphSAGE, Transolver, UPT), and classic UDA techniques (CMD, DeepCORAL, DANN), with evaluation across multiple model selection strategies.

**Dataset Code Accessibility:**

Yes

**Ethical Considerations:**

No, there are no or only very minor ethics concerns

**Final Justification:**

The author addressed my concerns. I will be increasing my rate.

**Limitations Weaknesses:**

1. Lack of Real-World Experimental or Sensor Data. Despite claiming industrial relevance, all tasks are based solely on synthetic simulation data generated from FEM or CFD solvers.

2. Evaluation Metrics Are Physically Shallow.  As seen in Table 2, performance is evaluated purely using scalar RMSEs on raw fields like deformation, stress, and temperature. However, these do not reflect physics consistency (e.g., residuals of PDEs, boundary condition violations, conservation errors).

3. Synthetic Shifts Are Limited to Scalar Parametric Gaps. All domain shifts are univariate (see Table 7), focused on one parameter per task. While this simplifies benchmarking, it overlooks more realistic or challenging shifts.

4. Consider including visualizations (e.g., error heatmaps or failure localization) to complement RMSE summaries.

5. UPT is only used on the heatsink task—clarify whether this is due to scalability constraints or mesh constraints.

6. Some results (e.g., Transolver rolling) appear unstable or anomalously large. Transolver on rolling has extreme errors (e.g., deformation: –579.11 mm, stress: –6409.53 MPa).  are there other physic-contstrained domain adaptation methods for this?

**Strengths Contributions:**

1. Bridging UDA and surrogate modeling for PDEs is an underexplored but critical area, particularly for simulation-driven design and digital twin applications.

2. Thoroughness: The authors benchmark 39 configurations per dataset (~1200 training runs total), conduct careful ablations, and report multiple seeds with error bars.

---

> ### Author Rebuttal · Authors · 2025-07-31
>
> We thank reviewer oE7g for their extensive and valuable review, which pinpoints important aspects we did not consider in the submission. Please find our answers below.
>
> ### 1. Lack of Real-World Experimental or Sensor Data.
> We agree that incorporating experimental or real-world data is an important aspect for digital twin applications. However, our benchmark arises from simulation-driven design optimization workflows, where FEM and FVM solvers remain the dominant source of ground truth data.
>
> Bridging the domain shift between simulation and real data is a key research direction. Nevertheless, __we believe it is orthogonal to our work__, as it would require different baseline algorithms (our UDA baselines rely on the same input space) and benchmarks (e.g., for sensor calibration, noise modeling, or hybrid inference). As research in sim2real for physical simulations has been very prolific in recent years, this is a very valid point to be added to the future work section and extensions to our benchmark.
>
> ### 2. Evaluation Metrics Are Physically Shallow.
> We agree that it's important to include more interpretable error metrics to make judgement of the surrogates easier. To address this, our domain expert co-authors suggested the following __application specific metrics__ they use in their industry projects. The metrics reflect how simulation outputs are usually validated in deployed industrial workflows.
>
> In our settings, engineers typically inspect the distribution of key physical quantities along a cord in the mesh. We present quantitative performance through average relative errors along these cords (source/target test sets), as an interpretable summary. The results are averaged over 4 seeds, numbers in parentheses indicate improvements compared to the unregularized baseline models (* means the unregularized models were unstable on target).
>
> - **Rolling:** Equivalent plastic strain PEEQ distribution along the vertical center cord of the slab.
>
> |Model|Best UDA method|Best model selection|Avg relative error|
> |---|---|---|---|
> |PointNet|CMD|SB|0.26 (–0.42)|
> |GraphSAGE|CMD|IWV|0.17 (–0.02)|
> |Transolver|CMD|SB|0.59 (*)|
> |Oracle (GraphSAGE)|Deep Coral|TB|0.16 (–0.03)|
>
> - **Forming:** Transverse stress (xx-component) distribution along the vertical cord at $x=\frac{l}{2}$.
>
> |Model|Best UDA method|Best model selection|Avg relative error|
> |---|---|---|---|
> |PointNet|Deep Coral|SB|1.15 (+0.27)|
> |GraphSAGE| DANN | IWV |5.60 (+3.10)|
> |Transolver| Deep Coral |DEV|0.80(+0.19)|
> |Oracle (Transolver)|CMD|TB|0.79 (+0.18)|
>
>
> - **Motor:** Von Mises stress distribution along the radial path starting from the right upper corner of the magnet pocket.
> The exact horizontal location can be calculated with an offset of  $\frac{t_{rsb1}}{2} + 1.1 * r_{r2}$ from the upper right corner of the domain.
>
> |Model|Best UDA method|Best model selection|Avg relative error|
> |---|---|---|---|
> |PointNet|Deep Coral|SB|0.20 (−0.15)|
> |GraphSAGE|CMD|SB|0.33 (−0.10)|
> |Transolver|Deep Coral|SB|0.09 (−0.02)|
> |Oracle (Transolver)|Deep Coral|TB|0.09 (−0.02)|
>
> - **Heatsink:** Temperature distribution along a cord along the width of the domain at the half slice of the length and a height of $y=0.025m$.
>
> |Model|Best UDA method|Best model selection|Avg relative error|
> |---|---|---|---|
> |PointNet|Deep Coral|SB|0.03 (−0.020)|
> |Transolver|Deep Coral|IWV|0.01 (−0.001)|
> |UPT|Deep Coral|SB|0.01 (+0.001)|
> |Oracle (UPT)|Deep Coral|TB|0.01 (−0.001)|
>
> ### 3a. Synthetic Shifts Are Limited to Scalar Parametric Gaps
> This is an excellent point, also raised by reviewer ErFp.
>
> Effectively, in 3 of the 4 presented datasets, __the domain shift defining parameter(s) describe the geometry of the samples__ (see Table 7 together with Figures 3a, 11 and 12); meaning that the geometry is parametrized by a set of scalars.
>
> The idea behind our conditioning setup is to allow plug-in integration of any model architecture and domain adaptation method.
>
> In initial ablations of our benchmark we experimented with mesh encoder conditioning networks instead of explicitly conditioning on the scalar parameters. It is worth noting that this choice is not architecture agnostic.
>
> Below, we include results on the motor dataset of a PointNet architecture with a geometric conditioner and without explicit access to the scalar parameters. We follow the procedure described in our paper: we train the model with each UDA algorithm and 9 different regularizer strengths, over 4 seeds (mean shown in results below). Values in the parentheses show improvement to the unregularized models.
>
> | UDA method | Best model selection | All fields normalized averaged (-) | Mises Stress (MPa) |
> | --- | --- | ---- | ---- |
> | DANN | SB | 0.40 (−0.07) | 32.96 (−3.81) |
> | CMD | SB | 0.39 (−0.07) | 32.79 (−3.99) |
> | Deep Coral | SB | 0.38 (−0.09) | 31.25 (−5.53) |
> | Oracle (Deep Coral) | TB | 0.38 (−0.09) | 31.25 (−5.53) |
>
> While UDA consistently improves target error, the errors are higher compared to our proposed benchmark setup (see Mises Stress in Table 2 and all fields normalized avg Table 5). Because of worse performance and less trivial extension to other architectures, we decided to go for the conditioning MLP in our final design (see Figure 1).
>
> ### 3b. All domain shifts are univariate (see Table 7).
> Thanks for raising this important remark, also noted by reviewer ErFp.
>
> We agree that multi-dimensional shifts present a more challenging scenario. To address this, we add preliminary results on __2-dimensional shifts__ in the electric motor design dataset. In more detail, we jointly shift the rotor slot diameter $d_{r3}$ (parameter shifted in the cases from the paper) and the angle between the magnets $\alpha$. We split the domains as follows:
>
> | Parameter | Source range | Target range |
> | --- | --- | --- |
> | Rotor slot diameter $d_{r3}$ | [99, 120) | [120, 130] |
> | Angle between magnets $\alpha$ | [119, 153) | [153, 170]
>
> We trained 336 additional models on this configuration, and present a summary of the results in the table below:
>
> | All Models | Best UDA method | Best model selection | All fields normalized avg (-) | Mises Stress (MPa) |
> | --- | --- | --- | --- | --- |
> | PointNet | DANN | SB | 0.35 (−0.09) | 28.84 (−5.02) |
> | GraphSAGE | Deep Coral | IWV | 0.33 (−0.05) | 28.27 (−1.39) |
> | Transolver | DANN | SB | 0.11 (−0.04) |  8.36 (−2.01) |
> | Oracle (Transolver) | CMD | TB | 0.11 (−0.04) |  8.30 (−2.06) |
>
> The table shows the best performing UDA algorithm and unsupervised model selection combination per architecture. Again, we report mean error values on the target domain accross 4 seeds and the improvement towards the corresponding unregularized baseline models in parentheses. Comparing these results with the original 1-dimensional shift (Table 2), we highlight two aspects:
>
> - __Increased error__: for 3 out of 4 architectures, the average error in Mises Stress is consistently larger than in the 1D shift setting, confirming that the 2D shift is indeed more challenging.
> - __Greater benefits from UDA__: the (relative) improvements compared to the unregularized baseline suggest that domain adaptation training is more beneficial, for this particular case.
>
> ### 4. Consider including visualizations (e.g., error heatmaps or failure localization) to complement RMSE summaries.
> We agree that visualizing spatial error patterns can offer valuable insights into model performance beyond scalar RMSE metrics. However, for the proposed datasets it is not trivial to visualize quantitative error heatmaps across whole testsets since they are represented as irregular meshes. The number and positions of mesh points, and therefore the geometry, varies within each dataset. An option is to map every mesh to a common reference grid, but this would introduce interpolation artifacts that obscure results instead of clarifying them.
>
> That said, we would like to direct reviewer oE7g to __Section 3 in the supplementary material__, where we provide representative visualizations of the worst and best predicted test samples for each dataset, both in the source and target domains.
>
> ### 5. UPT is only used on the heatsink task—clarify whether this is due to scalability constraints or mesh constraints.
> In Section 4.4. (Paragraph 3), it is mentioned that we chose to only apply UPT on the heatsink dataset since UPT uses a latent field representation that does not natively handle mesh deformations. Contrary to the heatsink dataset, the other three datasets involve mesh deformations, making UPT unsuitable without significant architectural alterations.
>
> ### 6a. Some results (e.g., Transolver rolling) appear unstable or anomalously large.
> We appreciate reviewer oE7g for noticing this. Indeed, some Transolver runs on the rolling dataset are unstable in the target domain (denoted by asterisks in the Appendix A.1, Table 3). To clarify, these instabilities only arise in the target domain, where no labels are available for supervised training, making them effectively undetectable at source-best model selection. We also want to point out that the unregularized runs are unstable. However, the best UDA algorithm for this case __(CMD) appears to stabilize target performance.__ We consider this fact a clue for the strengths and potential of using UDA in neural surrogate training.
>
> ### 6b. Are there other physics-constrained domain adaptation methods for this?
> Most of domain adaptation works have been in simple classification. In regression, there has been significantly less work. This is further limited when dealing high-dimensional regression, like it is the case for our datasets. As far as we know, there has been no or extremely sparse research explicitly done into general physics-constrained unsupervised domain adaptation for PDE surrogate modeling. This is one of our main motivations that led us to develop this benchmark. A specific physics-informed adaptation method is a natural follow-up.

---

> > ### Comment · Reviewer_oE7g · 2025-08-08
> >
> > Thank you very much for your detailed response. Please incorporate the points above into the revised paper. I will be increasing my rating to a weak accept.

---

### Comment · Area_Chair_NuHJ · 2025-08-04

Dear authors,

I am the AC here. I noticed that most reviewers have moderate confidence. To give you more opportunities to engage in the review process, I went through the review and paper by myself. It would be great if you could respond to the following high-level questions:

1. What is the importance of distribution shift in neural surrogates and industrial simulation tasks? Clearly, AI models can experience a performance drop when the deployment environment changes, including the neural surrogates-related problem. For industrial simulation tasks, can you provide a concrete scenario in which such a harmful shift would be significantly concerning?

2. Designing the shifts. Reviewers proposed a common question about how the shifts happen. For example, *synthetic shifts are limited to scalar parametric gaps. All domain shifts are univariate (see Table 7) and focus on one parameter per task*. What is the rationale behind doing so in your current paper?

3. Several reviewers had questions about time-dependent PDEs. In the industrial (simulation) tasks that you describe, such as the design problem, my question is: In general, are steady-state or time-averaged solutions sufficient? Is considering time-dependent PDEs a good complementary element, but not necessarily an essential one, in the relevant design problems? Do you have any supporting references?

---

> ### Author Response · Authors · 2025-08-05
> **Reply to Area Chair NuHJ**
>
> It was admittedly slightly frustrating to see such low confidences in reviews and not receive any replies to our rebuttal. We sincerely appreciate the area chair for noticing this and stepping in. Thanks for your time spent reading the paper as well as the reviews. We address all additional questions below.
> ### ad 1.)
> Neural surrogate models are increasingly deployed in industrial contexts to approximate expensive numerical simulations (e.g. FEM, FVM) in two scenarios:
> - For __design exploration__, where surrogates enable rapid evaluation of new designs under varying parameters.
> - For production monitoring or __digital twin applications__, where surrogates can provide fast, real-time estimates of moitoring quantities in the presence of noisy or shifting inputs.
>
> In both cases, inputs encountered at deployment may fall outside the training distribution (e.g. novel geometries, new materials, measurement drift, etc.). As an example: in automotive crashworthiness simulations, a neural surrogate trained on standard vehicles may misjudge a new lightweight design (thinner beams, composite materials), causing inaccurate safety assessments. Domain adaptation can help reduce such extrapolation errors.
>
> Moreover, this concern is not only hypothetical. The importance of **robustness of deployed AI systems is now also reflected in regulatory bodies** such as the **EU AI Act** (Article 15) [1]. Ensuring that models remain robust under domain shift is thus not only a technical necessity, but also a compliance issue.
> ### ad 2.)
> In industrial design workflows, simulation problems are typically parameterized, and the design process is driven by an outer optimization algorithm [2]. These algorithms explore the design space by varying the input parameters and then evaluate configurations with a fast surrogate.
>
> This __practical setup motivates our definition of distribution shifts:__ we define them explicitly in the parametric input space, as this is more controllable than implicit shifts in data space, which are harder to define and quantify. Still, data space shifts can be a useful addition, as noted by reviewer 2uJ6; we addressed this in our rebuttal (Table in rebuttal 2uJ6, point 2).
>
> We agree that univariate (1D) shifts are an idealized setting. Reviewer also oE7g pointed this out, and we __added results on 2D shifts__ (review oE7g, 3b). They show that methods effective in 1D also generalize to multivariate shifts. We intentionally start with 1D, as SIMSHIFT is, to our knowledge, the first benchmark systematically evaluating UDA methods for high-dimensional regression, rather than the usual classification setting.
>
> We hope that this benchmark can serve as a foundation fo the development of specialized UDA methods tailored to such settings.
> ### ad 3.)
> Most industrial design optimization tasks rely on either steady-state or time-averaged solutions rather than detailed transient dynamics. This is not just a modeling convenience, but reflects how simulation is integrated into design pipelines.
>
> In initial design stages, numerical simulations are used to assess candidate designs by computing scalar objective values such as drag and lift coefficients, maximum stresses, or thermal loads. These quantities can typically be obtained from steady-state solvers or through statistical aggregation of transient simulations, e.g. by solving Reynolds-averaged Navier Stokes (RANS) equations or large eddy simulations (LES). This practice is well documented across various application areas, including thermal systems [3], aerodynamic shape optimization for aircrafts [4, 5], wind turbine design [5], and car aerodynamics [2].
>
> Further, __we deliberately focus on steady-state problems to reflect those standard settings__. While for some of our presented datasets, transient modeling does not make sense physically (e.g. motor), for others it would simply not provide more useful information for the target design tasks (e.g. heatsink).
>
> We acknowledge that transient dynamics are crucial in other applications, with probably the most renowned one being weather modeling. These domains involve different challenges, including temporal error accumulation (as mentioned in our paper and review 2uJ6, point 1), and we view the inclusion of time-dependent PDEs as an extension of our work.
> ## References
> [1] *EU Artificial Intelligence Act*, Chapter III, Article 15.
>
> [2] L. Dumas, *CFD-based optimization for automotive aerodynamics*, Optimization and Computational Fluid Dynamics, D. Thévenin and G. Janiga, Eds. Berlin, Heidelberg: Springer, 2008, pp. 191–215.
>
> [3] P. Majumdar, *Design of Thermal Energy Systems*, Chapter 9: Simulation of Thermal Systems. Wiley, 2021.
>
> [4] S. N. Skinner and H. Zare-Behtash, *State-of-the-art in aerodynamic shape optimisation methods*, Applied Soft Computing, vol. 62, pp. 933–962, 2018.
>
> [5] J. R. R. A. Martins, *Aerodynamic design optimization: Challenges and perspectives*, Computers & Fluids, vol. 239, 105391, 2022.

---

### Note · Authors · 2025-08-12

We appreciate the constructive feedback from all the reviewers and the AC, recognizing the thoroughness of our experimental setup and the novelty and importance of UDA in neural surrogates.

We summarize our contributions as follows:
- SIMSHIFT is the first application of Unsupervised Domain Adaptation to neural surrogates. This is a critical area for industrial simulations, where performance and robustness are not only required by the design task, but mandated by legislation.
- We propose 4 novel industry-relevant simulation datasets, spanning from metal manufacturing, to machinery and electronics. Each dataset is packaged with domain shifts in mind, providing different domain splits of increasing difficulty.
- We adapt and benchmark multiple UDA methods and neural surrogate models, resulting in 1200 total runs.

Additionally, we noticed two recurrent concerns:
1. __Nature of the shifts__: we addressed this in the rebuttal, by providing preliminary results on both _multivariate_ (2D)  and _geometric_ (non-parametric) cases, showing consistent performance and showcasing the flexibility of our benchmark.
2. __Time dependent systems__:  we reckon most simulations in industry are not transient (steady-state or time averaged). SIMSHIFT datasets reflect this, as transient modeling is not physically meaningful (electric motor), or it does not provide useful information for the target design tasks (heatsink). We view time-dependent systems as a valuable extension of our work, in particular for large-scale domains such as weather modeling (where shifts are frequent).

Thank you again for your time and effort!

The Authors

---

### Decision · Program_Chairs · 2025-09-18

**Decision:**

Reject

**Comment:**

This paper presents SIMSHIFT, a benchmark suite for Unsupervised Domain Adaptation (UDA) applied to neural PDE surrogates across four industrial simulation tasks: hot rolling, sheet metal forming, electric motor design, and heatsink design. The benchmark includes parametric domain shifts, baseline neural operators (PointNet, GraphSAGE, Transolver, UPT), and classic UDA techniques (CMD, DeepCORAL, DANN), with evaluation across multiple model selection strategies.

Indeed, this paper is borderline, with most reviewers providing moderate reviews and confidence. I understand this approach, given that the paper covers different areas: neural surrogates for PDEs, distribution shifts, and industrial simulation tasks. This combination makes the paper more difficult to follow and evaluate in terms of its actual contributions. Based on this, I initiated a discussion with the authors regarding the key disputes. The following important factors were considered when making recommendations.

[1] Importance of distribution shift in neural surrogates and industrial simulation tasks.

> Neural surrogate models are increasingly deployed in industrial contexts to approximate expensive numerical simulations (e.g. FEM, FVM) …..

> In both cases, inputs encountered at deployment may fall outside the training distribution (e.g. novel geometries, new materials, measurement drift, etc.). As an example: in automotive crashworthiness simulations, a neural surrogate trained on standard vehicles may misjudge a new lightweight design (thinner beams, composite materials), causing inaccurate safety assessments. Domain adaptation can help reduce such extrapolation errors.

> Moreover, this concern is not only hypothetical. The importance of robustness of deployed AI systems is now also reflected in regulatory bodies such as the EU AI Act (Article 15) [1]. Ensuring that models remain robust under domain shift is thus not only a technical necessity, but also a compliance issue.

I am satisfied with the authors' concrete responses.  Specifically, AI-based neural surrogate models are emerging in industrial contexts to approximate expensive numerical simulations, with strong potential applications. The authors also replied regarding the practical and legal requirements for considering distribution shift scenarios.

[2] Designing the shifts

> In industrial design workflows, simulation problems are typically parameterized, and the design process is driven by an outer optimization algorithm [2]. These algorithms explore the design space by varying the input parameters and then evaluate configurations with a fast surrogate.

> This practical setup motivates our definition of distribution shifts: we define them explicitly in the parametric input space, as this is more controllable than implicit shifts in data space, which are harder to define and quantify. Still, data space shifts can be a useful addition, as noted by reviewer 2uJ6; we addressed this in our rebuttal (Table in rebuttal 2uJ6, point 2).

> We agree that univariate (1D) shifts are an idealized setting. Reviewer also oE7g pointed this out, and we added results on 2D shifts (review oE7g, 3b). They show that methods effective in 1D also generalize to multivariate shifts. We intentionally start with 1D, as SIMSHIFT is, to our knowledge, the first benchmark systematically evaluating UDA methods for high-dimensional regression, rather than the usual classification setting.

Based on the nature of the shift and additional experiments in 2D parameters. I am convinced that authors addressed this.


[3] About time-dependent PDEs.

> Most industrial design optimization tasks rely on either steady-state or time-averaged solutions rather than detailed transient dynamics. This is not just a modeling convenience, but reflects how simulation is integrated into design pipelines.

> In initial design stages, numerical simulations are used to assess candidate designs by computing scalar objective values such as drag and lift coefficients, maximum stresses, or thermal loads. These quantities can typically be obtained from steady-state solvers or through statistical aggregation of transient simulations, e.g. by solving Reynolds-averaged Navier Stokes (RANS) equations or large eddy simulations (LES). This practice is well documented across various application areas, including thermal systems [3], aerodynamic shape optimization for aircrafts [4, 5], wind turbine design [5], and car aerodynamics [2].

> Further, we deliberately focus on steady-state problems to reflect those standard settings. While for some of our presented datasets, transient modeling does not make sense physically (e.g. motor), for others it would simply not provide more useful information for the target design tasks (e.g. heatsink).

Based on the authors' responses, I am convinced that steady-state or time-averaged solutions are sufficient for most applications relevant to the industry.

Therefore, regarding the technical concerns within [2–3], I would think that the authors addressed them reasonably well, and I would recommend that they include them in the final version.

**Rationale for the final recommendation: I believe this paper proposes a new perspective on real-world distribution shift settings. Distribution shifts are often observed in certain applications, such as healthcare, computer vision and NLP. However, the implications of distribution shifts in other real-world applications remain unclear. Unlike other common practical scenarios, this paper focuses on shifts in scientific computing and their further applications in industrial design. This makes the paper difficult to evaluate under regular scenarios, such as those in healthcare. Nevertheless, I believe NeurIPS should have a place for this one.**

===== FINAL UPDATE FROM DB Track PCs ====

The final decision for this paper has been taken by the program chairs after consultation with the SACs. All Senior Area Chairs have ranked papers according to the feedback from the AC during the review process. We decided to leave the original meta-review to reflect the opinion of the AC in light of the initial discussions with reviewers and SAC.